
# Surface impacts of the Quasi Biennial Oscillation

Lesley J. Gray[1], James A. Anstey[2], Yoshio Kawatani[3], Hua Lu[4], Scott Osprey[1], Verena Schenzinger[1,5]

[1]National Centre for Atmospheric Sciences (NCAS), Department of Ocean, Atmosphere and Planetary Physics, University of Oxford, Oxford, OX1 3PU, U.K.

[2]Canadian Centre for Climate Modelling and Analysis, Environment and Climate Change Canada, University of Victoria, Victoria, V8W 2Y2, Canada

[3]Japan Agency for Marine-Earth Science and Technology, Yokohama, 236-0001, Japan

[4]British Antarctic Survey, Cambridge, CB3 0ET, U.K.

[5]Institute for Meteorology and Geophysics, University of Vienna, 1090 Vienna, Austria.

*Correspondence to*: Lesley J. Gray (lesley.gray@physics.ox.ac.uk)

**Abstract.** Teleconnections between the Quasi Biennial Oscillation (QBO) and the Northern Hemisphere zonally-averaged zonal winds, mean sea level pressure (mslp) and tropical precipitation are explored using regression analysis. A novel technique is introduced to separate responses associated with the stratospheric polar vortex from other underlying

mechanisms. A previously reported mslp response in January, with a pattern that resembles the positive phase of the North Atlantic Oscillation (NAO) under QBO westerly conditions, is confirmed and found to be primarily associated with a QBO modulation of the stratospheric polar vortex. This mid-winter response is relatively insensitive to the exact height of the maximum QBO westerlies and a maximum response occurs with westerlies over a relatively deep range between 10-70hPa. Two additional mslp responses are reported, in early winter (December) and late winter (February / March). In contrast to

the January response the early and late winter responses show maximum sensitivity to the QBO winds at ~20 hPa and ~70 hPa but are relatively insensitive to the QBO winds in between (~50 hPa). The late winter response is centred over the North Pacific and is associated with QBO influence from the lowermost stratosphere at tropical / subtropical latitudes. The early winter response consists of anomalies over both the North Pacific and Europe, but the mechanism is unclear and requires further investigation. QBO anomalies are found in tropical precipitation amounts and a southward shift of the Inter-

tropical Convergence Zone under westerly QBO conditions is also evident.

## 1 Introduction

A modulation of the winter Northern Hemisphere (NH) stratospheric polar vortex by the equatorial Quasi Biennial Oscillation (QBO) has been well known for many years (Holton and Tan 1980, 1982, Baldwin et al. 2001, Anstey and Shepherd 2014). Potential mechanisms for this influence have been extensively explored. Modelling and observational

studies suggest it is related to the influence of the equatorial winds $U_{eq}$ on the stratospheric waveguide, which in turn affects the vertically propagating planetary-scale waves from the troposphere. Nevertheless, the exact mechanisms are still under debate (Garfinkel et al. 2012; Watson and Gray, 2014; White et al 2016). It is also evident that the Holton-Tan relationship



between the QBO and NH polar vortex, with a stronger (weaker) polar vortex when the QBO winds in the lower-mid stratosphere are westerly (easterly) is more evident in some periods than in others (Lu et al. 2008, 2014; Anstey and Shepherd 2008; Christiansen 2010). The majority of climate models currently employed to assess the impacts of climate change do not reproduce the QBO, and even those that do are often unable to reproduce the observed strength of the Holton-

Tan correlation (Christiansen, 2014).

Variability of the stratospheric polar vortex is not only influenced by the QBO. Signals associated with the occurrence of explosive volcanic eruptions (Robock 2000), ENSO events (Richter et al. 2011) and the 11-year solar cycle (Gray et al. 2010) have all been identified. There is, in addition, a strong influence of natural variability, both internally within the

stratosphere and within the coupled ocean – atmosphere – land surface system that impacts the amplitude and propagation of vertically propagating waves from the troposphere. Because of this, there is currently limited predictability of, for example, sudden stratospheric warming events in the stratosphere.

The mechanism for stratospheric processes to influence the underlying troposphere and surface is currently not well

understood, neither in the extra-tropics where polar vortex variability is the prime candidate (Kidston et al. 2015) nor at equatorial / subtropical latitudes where the direct impact of the QBO may be more relevant. Nevertheless, a large number of studies have demonstrated impact of stratospheric variability on the underlying tropospheric circulation, especially in the Atlantic sector (Baldwin and Dunkerton 2001; Thompson et al. 2002; Scaife et al. 2014, 2016; Kidston et al. 2015; Hansen et al. 2017). An improved ability to simulate stratospheric processes and their impact on surface weather and climate has the

potential to improve seasonal and decadal-scale forecasts of surface weather patterns such as the North Atlantic Oscillation (NAO; Marshall and Scaife 2009; Scaife et al. 2016). Similarly, the direct impact of the QBO on the underlying tropospheric circulation in the tropics and sub-tropics is of interest (Giorgetta et al. 1999; Collimore et al. 2003; Ho et al. 2009; Liess and Geller 2012; Nie and Sobel 2015) and recent studies have shown that the QBO modulates the amplitude of the Madden Julian Oscillation, which may lead to improved predictability of MJO under QBO-E conditions (Yoo and Son 2016;

Marshall et al. 2016; Son et al. 2017; Nishimoto and Yoden 2017).

One mechanism for QBO influence at the surface is through the modulation of planetary wave propagation in winter by refraction or horizontal reflection of the waves towards (away from) high latitudes under QBO-E (QBO-W) conditions (Holton and Tan 1980; 1982; Garfinkel et al. 2012; Watson and Gray 2014; White et al. 2015). This influence route (the

'polar route') modulates the strength of the polar vortex, and there have been many studies demonstrating that the polar vortex can subsequently influence the underlying tropospheric circulation and surface temperature / pressure distributions (Baldwin and Dunkerton 2001; Thompson et al. 2002, Mitchell et al. 2013). Changes in the mid-upper stratospheric zonal flow (and particularly the vertical and/or horizontal potential vorticity gradients) can also result in vertically reflected waves





and these can influence the surface without the direct mediation of a substantial change (or reversal) in the strength of the polar vortex (Perlwitz and Harnik 2003, 2004; Shaw and Perlwitz 2013; Lu et al. 2017).

The QBO also influences the temperature in the tropical and subtropical lower stratosphere, as a result of the induced
adiabatic meridional circulation required to maintain thermal wind balance (the 'subtropical route'). This influences wind shears in the vicinity of the tropospheric subtropical jet, which in turn affects the growth and life-cycle of mid-latitude synoptic-scale (baroclinic) and planetary-scale waves in the troposphere (Ruti et al. 2006; Simpson et al. 2009; Garfinkel and Hartmann 2011). Finally, the QBO directly influences the vertical wind shear in the tropics thus influencing, for example, tropical tropopause temperatures and static stability (the 'tropical route'). There is both modelling and observational
evidence for a QBO signal in the characteristics of deep tropical convection and precipitation (Giorgetta et al. 1999; Collimore et al. 2003; Ho et al. 2009; Liess and Geller 2012; Son et al. 2017).

These three possible routes for QBO influence at the surface (polar, subtropical, tropical) are unlikely to act in isolation, making it difficult to distinguish between them. For example, a QBO influence on the polar vortex will impact the surface
not only at mid-to-high latitudes via the polar route but also potentially in the tropics and subtropics via its impact on the strength of the upwelling branch of the Brewer-Dobson circulation, which can then influence the tropical troposphere via the tropical route. Conversely, QBO-related changes in tropical convection via the tropical route influences equatorial wave generation which feeds back on to the QBO and also affects the source of stationary Rossby waves that propagate horizontally away from the tropics and may thus affect both the subtropical and the eddy-driven jet (Hoskins and Karoly
1981; Scaife et al. 2017). In this way a feedback occurs both within the tropics and also to higher latitudes.

In summary, there are multiple possible influence and feedback routes (teleconnections) between the QBO and the surface. While a data study alone is unable to demonstrate cause and effect, nor isolate these individual routes of influence and feedbacks, it can nevertheless point to the major potential contributors and provide hypotheses for climate model
experiments to test.

In data studies that seek to investigate the influence of the QBO, a single time-series index to characterise the state of the QBO is desirable. The standard approach is to use the zonally-averaged equatorial zonal wind $U_{eq}$ at a selected level such as 40 or 50 hPa.  However, this has the drawback of characterising the behaviour at only one level and cannot provide
information on the influence of the QBO winds from other levels; the selected level is often chosen on a trial-and-error basis to find the 'best' level to employ, which has resulted in studies using a variety of different levels between 20-70 hPa. A more objective approach employs Empirical Orthogonal Function (EOF) analysis of the equatorial winds within a selected height range so that the height dependence of the QBO can be also characterised (Wallace et al. 1993). This has been successfully



employed in various previous studies (e.g. Baldwin et al. 1998 hereafter denoted BD98; Crooks and Gray 2005; Anstey et al. 2010, Rao and Ren 2017).

In this study, we follow the approach of BD98 who characterised the QBO using EOF analysis to derive a single QBO time-
series and used it to explore the QBO influence on the NH stratospheric polar vortex. Their study is extended in several ways. The analysis period is increased from 1978-1996 (18 winters) to 1958-2016 (58 winters) and regression analysis is performed instead of composite analysis so that other sources of variability are more effectively accounted for. The scope of the analysis is widened so that it examines not only QBO-related variability of the stratospheric polar vortex winds but also QBO variability in the troposphere and at the Earth's surface in both mid-latitude and tropical latitudes. This involves
analysis of mean sea level pressure (mslp) and tropical precipitation, in addition to zonally-averaged zonal wind fields. Finally, a novel approach is explored that aims to separate out the role of the polar vortex (i.e. the polar route) so that the remaining QBO variability in the troposphere can be more effectively identified and analysed.

The paper is organised as follows: section 2 describes the datasets employed and the analysis approach, including a detailed
description of the EOF analysis to determine the optimum QBO anomaly profile. Section 3 describes and discusses the results of the analyses. Firstly, the polar vortex response is explored and compared with the results of BD98. A closer examination is then made of the tropospheric response in the zonally averaged zonal winds. The same techniques are then applied to the mslp and precipitation fields. Section 4 provides a summary of the results and their implications.

**2 Datasets and Methodology**

Regression analyses are performed for the period 1958-2016 on zonal wind and precipitation fields from the European Centre for Medium Range Weather Forecasting (ECMWF), namely the ERA-40 (1958-1979) and ERA-Interim (1979-2016) datasets and on mean sea level pressure fields from the Hadley Centre HadSLP2 dataset. The focus of the study is on the NH surface response.

The monthly-averaged, zonally-averaged and 3-dimensional fields from ERA-40 (Uppala et al. 2005) and ERA-Interim (Dee et al. 2011) were obtained via the ECMWF public dataset web access (http://apps.ecmwf.int/datasets/). The extracted fields were those interpolated on to standard pressure levels and 2.5 degrees horizontal resolution. While we note the presence of jumps in the reanalysis fields due to the introduction of additional satellite data and the use of parallel processing streams (e.g. Long et al. 2017), these are primarily evident in the temperature fields and are much less evident in the zonal wind
fields. The two datasets overlap in the period 1979-2001; we chose to employ the more up-to-date data from ERA-Interim for this period but tests showed that the results were not sensitive to this choice. The precipitation fields were only analysed in the period 1979-2016 when the more reliable ERA-Interim fields were available.





The mslp data are from the Hadley Centre HadSLP2 dataset (Allan and Ansell, 2006) for the period 1958-2004 together with the more recent data since 2004 from the HadSLP2r dataset. The dataset comprises observations across the whole globe from 2227 land stations, with 615 providing data continuously for more than 100 years. These are complemented by marine

observations from the International Comprehensive Ocean Atmosphere Data Set (Worley et al. 2005) and the Marine Data Bank from the UK Met Office. The data have been quality controlled, gridded, interpolated and blended to produce a dataset of monthly-averaged mslp with a spatial resolution of 5°.

The multivariate linear regression analysis is described in Gray et al. (2013; 2016): the time-series at each grid-point is fitted

using a number of indices (time-series) that characterise the observed variability associated with (1) volcanic eruptions, (2) El Nino Southern Oscillation (ENSO); (3) solar radiative forcing, (4) the QBO and (5) a long-term trend. The GISS updated Sato Index (Sato et al. 1993) is used as the volcanic index. ENSO variability is characterised by a time-series of averaged sea surface temperatures (SSTs) from the Nino 3.4 region (120-170°W, 5°N-5°S) using monthly averaged data on a 1° spatial grid from the Hadley Centre HadISST dataset (Rayner et al. 2003; https://www.esrl.noaa.gov/psd/

gcos_wgsp/Timeseries/Nino34/). Solar variability is characterised using the time-series of sunspot numbers from WDC-SILSO (Royal Observatory of Belgium, Brussels; http://www.sidc.be/silso/). A simple linear trend is used for the long-term trend index. The various indices used to characterise the QBO are described below. The best fit to the normalised anomalies is calculated by minimising the residual term and statistical significances are determined using a Student's t-test with the null hypothesis that the contribution to the variability from a particular index is zero, taking into account the autocorrelation of

the time-series using a monthly AR(1) process. The resulting regression coefficients are then re-scaled by multiplying the coefficients by the ratio of the max-min amplitude of the index and its standard deviation.

Sensitivity tests showed that the QBO regression coefficient results in the following sections are unaffected by the inclusion or exclusion of the solar, ENSO and/or trend terms, confirming that these indices are independent of each other. The

sensitivity tests also confirmed that the data have not been over-fitted by inclusion of too many indices since the QBO regression coefficients from the multivariate regression with all five indices were essentially identical to the univariate regression when only the QBO index was included (variability associated with the other indices was transferred to the residual in this latter case). Finally, tests were carried out to confirm that results using only the ERA-Interim period since 1979 were essentially the same as those from the whole period 1958-2016.

The QBO index was derived from radiosonde observations issued by the Free University of Berlin (Naujokat, 1986; FUB 2016). The FUB data are a combination from three different stations: Canton Island (3°S/172°W; January 1953-August 1967), Gan / Maldive Islands (1°S/73°E; September 1967- December 1975) and Singapore (1°N/104°E; since January 1976). The merged data are provided as monthly averages interpolated on the 70, 50, 40, 30, 20, 15 and 10 hPa levels. The QBO





index was calculated from the FUB data rather than using the ERA equatorial winds so that it characterised the observed QBO as closely as possible, thus avoiding any possible degradation by the data assimilation process (Kawatani et al. 2016).

Empirical Orthogonal Function (EOF) analyses are performed to obtain a single time-series to represent the QBO (Wallace
et al. 1993). Figure S1 shows the height profile and principal component (PC) time-series of the first two EOFs of the monthly-averaged deseasonalised FUB equatorial wind time-series between 10-70 hPa for the period 1958-2016. The PC time-series $P_1$ and $P_2$ are oscillations with an approximate 2-year period and are ~90$^o$ out of phase. Together the two EOFs account for ~71% of the field variance. EOF$_1$ (39.7%) characterises the anti-correlation of the equatorial winds in the upper and lower stratosphere whereas EOF$_2$ (31.0%) captures the variability at the intermediate heights in the mid-stratosphere.
Note that the total explained percentage is substantially lower than the ~95% found e.g. by BD98 and Anstey and Shepherd (2014) because we do not smooth the data with a 5-month running mean.

Figure S2 shows the reconstructed equatorial zonal wind field using only the first two EOFs. The main structure of alternating easterly and westerly shear zones is represented well, including the difference in the descent rates of the easterly
and westerly shear zones, although some details such as the long stalling phases, where the descent of the easterly shear zone can be halted for up to 6 months (e.g. in 2000), are smoothed out in the reconstruction.

The QBO phase-space (Figure 1), shows the amplitude of $P_1$ (abscissa) versus $P_2$ (ordinate) for each month of the time-series, and forms an approximate circle around the origin. The phase angle of a particular point in time is defined as
$\psi$=arctan($P_2/P_1$). Also shown, are example profiles showing the height distribution for selected combinations of $P_1$ and $P_2$, at 30$^o$ intervals (Schenzinger 2017). These are calculated as cos($\psi$)$\mathbf{e}_1$ + sin($\psi$)$\mathbf{e}_2$ where $\mathbf{e}_1$ and $\mathbf{e}_2$ are the EOFs shown in figure S1. Note that at $\psi$=0$^o$ and $\psi$ =90$^o$ the profiles are identical to EOF-1 and EOF-2 respectively, while at $\psi$=180$^o$ and $\psi$=270$^o$ they are the mirror image. Examining the plot in an anti-clockwise direction (with $\psi$ increasing, starting from $\psi$=0$^o$), a westerly phase in the upper stratosphere can be observed to steadily descend to the lower stratosphere by $\psi$ =90$^o$. This is
gradually replaced by an easterly phase in the upper stratosphere by $\psi$=180$^o$. Likewise, this easterly phase descends to the lower stratosphere by $\psi$=270$^o$ and is replaced by a westerly phase at $\psi$=360$^o$, thus completing the cycle. Note that for the descending easterlies (60$^o$ < $\psi$ < 240$^o$) the phase space is more densely populated. This is consistent with the well-known slower descent of the easterly phase of the QBO, so that the observations progress more slowly through these phase angles.

The length of a QBO cycle can be defined straightforwardly from the EOF analysis as the time needed for the completion of a full $2\pi$ cycle in phase space. The value derived in this way gives a mean period ± one standard deviation of 28.0 ± 3.6 months, which compares very well with standard estimates based on wind reversals. Similarly, the time derivative $\delta\psi/\delta t$ can be used to assess the downward phase progression of the QBO. A least squares fit of the descent rate of the zero wind line





(km/month) as defined in Schenzinger et al. (2016) vs. phase progression rate δψ/δt (cycles/month) indicates that they are highly correlated ($R^2$ = 0.68 with >99% statistical significance using a Student's t-test).

Following BD98 the first two EOFs are combined to provide a single time-series to characterise the QBO variations, by
defining u* = r sin(ψ + φ) where $r^2 = P_1^2 + P_2^2$, ψ = arctan ($P_1/P_2$) and φ is an arbitrary phase shift that can be interpreted as a rotation in phase space, and thus a projection onto a coordinate system that has been rotated by − φ compared to the one spanned by the original EOFs. For φ = $0^o$, u* = rsin(ψ) is the projection of a point ($p_1$, $p_2$) onto EOF-2. Addition of φ to the phase angle ψ gives u* = rsin(ψ + φ), which is the projection onto a new axis, EOF-2*. This new axis can be interpreted as a combination of EOF-1 and EOF-2, so that EOF-2* = (sin(φ), cos(φ)) where the coordinate system is spanned by EOF-1 = (1,
0) and EOF-2 = (0, 1). As an illustration, Figure S3 shows a sample period of the resulting time-series u* for different phase angles φ. The figure also demonstrates that the phase shift φ can also be interpreted as a shift in time. Given that the average cycle (2π) lasts for about 28 months, a step of φ =$10^o$ is equivalent to a time shift of (10 x 28/360) months (~0.8 months). Note that a positive shift in φ denotes a negative shift in time (clockwise rotation in the figure 1 phase space).

A series of correlations is then performed between the time-series of u* with various values of φ and the FUB $U_{eq}$ time-series at various heights to find the maximum correlation, and thus relate u* (and hence the phase shift φ) back to a particular QBO level of maximum influence. Figure S4 shows the correlation coefficients for the original zonal wind time-series $U_{eq}$ at each height between 10-70 hPa against u* at each phase shift between -$180^o$ < φ <$180^o$. The maximum correlation coefficients
exceed 0.9 for all but the lowermost (70 hPa) level, so the identification of the equatorial zonal wind $U_{eq}$ at a certain level with u* at a specific phase shift φ appears to be well justified. Note that values of (a) φ = -$60^o$, (b) φ = $0^o$ and (c) φ = +$60^o$ can be broadly equated to using a QBO time-series index $U_{eq}$ at 20 hPa, 40 hPa and 70 hPa respectively.

### 3 Results

#### 3.1 The polar vortex response

As described in section 2, the standard approach that examines the difference between composite fields derived according to the phase of the equatorial QBO at a specified pressure level can be improved in two ways. Firstly by employing regression analyses that take account of variability associated with ENSO, volcanic eruptions, the 11-year solar cycle plus a linear trend in addition to the QBO, and secondly by employing an EOF approach to encapsulate the height variations of the QBO. A QBO index u* is defined by u* = r sin (ψ + φ) where $r^2 = P_1^2 + P_2^2$, $P_1$ and $P_2$ are the principal components of the first two
EOFs of the equatorial wind in the region 10-70 hPa, ψ=arctan ($P_2/P_1$) and φ is an arbitrary phase shift that can be interpreted as a rotation in phase space (see section 2 for further details). This single time-series is then employed as the QBO index in

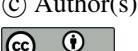



the regression analysis and the analysis is repeated with different values of ϕ in order to find the angle ϕ at which the response is maximised.

Figure 2 shows the scaled QBO regression coefficients of zonally-averaged zonal winds for November - March using values of (a) ϕ = -60°, (b) ϕ = 0° and (c) ϕ = +60° (which can be broadly equated to using a QBO time-series index $U_{eq}$ at 20 hPa, 40 hPa and 70 hPa respectively). All three have a westerly QBO phase anomaly in the mid-lower equatorial stratosphere (e.g. in the region 30-40 hPa) but each represents a slightly different stage of the progression of the descending phase difference (see figure 1). Scaling is applied so that the figures show typical amplitudes for QBO-W minus QBO-E differences (see section 2). A statistically significant response is evident over a relatively broad range of ϕ in each of the months, with maximum NH winter vortex response and greatest statistical significance in early winter in the range $20° > ϕ > -20°$ (hence our choice to show results at). These results, based on 58 years, agree remarkably well with BD98 (who analysed only 18 years) in terms of the optimum value of ϕ to characterise the maximum vortex response; their optimum ϕ response correlated most closely with the $U_{eq}$ time-series at ~45 hPa. Consultation with figure S4 shows that the u* time-series at ϕ =0° is correlated well with the equatorial QBO time-series at ~40 hPa. The general agreement of these results at ϕ = 0° with previous composite analyses is also unsurprising since most have utilised $U_{eq}$ at ~50 hPa to characterise the QBO.

These regression results thus confirm earlier studies (Baldwin et al. 2001; Anstey et al. 2010; 2014) that the primary influence of the QBO on the NH vortex is in early winter. There are some interesting similarities and differences between these results that employ regression analysis of the period 1958-2016 and the results of Anstey et al. (2010; hereafter A10) who employed composite analysis of only the ERA-40 dataset which covers the 1958-2001 period. The deep extension of the November signal into the NH troposphere is notable in both analyses. A10 note that a different EOF combination could be found to optimise an early winter response that peaked in November/December and a separate late winter response that peaked in February (see their figures 5 and 6). They found greatest statistical significance in November and this reduced as the winter progressed. In contrast, figure 2 indicates a maximum polar vortex response and greatest statistical significance in mid winter (January). The February response is much weaker and is statistically insignificant at all values of ϕ. This shows better agreement with the analysis of Anstey and Shepherd (2014) and Lu et al. (2014)who analysed the longer period 1958-2011 (see their figure 3a), suggesting that this difference is likely due to the additional years.

In the Southern Hemisphere (SH) the regression results also confirm the findings of BD98. The amplitude response of the SH vortex is found to be greatest (and statistically significant) in October / November. In contrast to the NH, this suggests that the QBO influence on the SH vortex is in late winter / spring; the QBO appears to primarily affect the timing of the final warming in the SH as opposed to the early/mid winter vortex strength in the NH. The SH vortex response maximises at approximately ϕ=30° (not shown) which is highly correlated with the $U_{eq}$ time-series at ~30hPa (see figure S4). This is a





slightly higher level than the $U_{eq}$ = 40 hPa level found to optimise the NH polar vortex response and is consistent with results of BD98 (and also A10 who analysed this hemispheric difference further).

At $\phi$=+60° (figure 2a) the regression coefficients show that when the QBO-W minus QBO-E difference is maximum in the
upper statosphere around 15-30 hPa i.e. the earlier stage in the descent of the anomaly, the main NH extra-tropical response is in the upper stratosphere, primarily over the subtropics in early winter. It moves poleward and extends downward throughout the course of the winter, although the response is only statistically significant in early winter. The NH subtropical response in December resembles the QBO-induced meridional circulation anomaly (opposite sign in the subtropics to that in the tropics; Plumb and Bell 1982), but the amplitude enhancement and poleward extension throughout the winter suggests
that it has been further amplified by wave-mean flow interaction, which is particularly sensitive to the background mean flow in this region (Gray et al. 2003). As an aside, we note the similarity between the evolution of this subtropical QBO feature in the upper stratosphere and the proposed mechanism for the response of the NH vortex to the 11-year solar cycle, which also involves the presence of a small early winter wind anomaly in the subtropical upper stratosphere that is amplified by wave – mean flow interaction and moves poleward and downward through the winter in a similar fashion (Kodera and
Kuroda 2002; Gray et al. 2010).

The QBO-induced circulation anomaly in the subtropics is more clearly evident in figure 3, which shows the corresponding QBO signal from a regression analysis in which an additional index was included to represent variations in the strength of the NH stratospheric polar vortex. The additional index consisted of the monthly time-series of zonally-averaged zonal
winds at 10 hPa, 60°N. Including indices that are not independent is inadvisable. The polar vortex has a QBO contribution to its variability and the QBO response is thus likely to change as a result of the inclusion of the vortex index. However, while acknowledging this issue, we choose to turn it to our advantage to explore the possibility that the resulting changes in the response patterns might help to separate out the different influence paths discussed earlier (polar, subtropical, tropical). The intention of this approach is therefore to remove, as far as possible, the response that is more closely associated with the
polar vortex variability than with the equatorial QBO variability. The remaining QBO response shown in figure 3 is thus the response that the regression analysis determines to be more closely associated with the equatorial QBO wind index than with the polar vortex index.

Despite the known multi-colinearity in the linear regression model between the QBO and the polar vortex, the presence of
the multi-colinearity does not appear to adversely affect the efficacy of the fitting process. Various sensitivity tests have been performed to check the efficacy of this approach. Firstly, we confirmed that the QBO results with and without the additional vortex index were not influenced by inclusion / exclusion of any of the other indices. Secondly, a stepwise approach was taken, in which variability associated with the polar vortex was first regressed out (using a univariate regression with just the vortex index) and then the standard multi-variate analysis (with QBO, ENSO, solar, volcanic, trend terms) was performed on



the residual. Comparison of the results with the single-step process that included all indices at once found them to be essentially the same.

As expected, the polar vortex response previously associated with the QBO has disappeared in figure 3 (e.g. compare figure
3b with figure 2b in January), since this is better captured by the vortex index. The induced meridional QBO circulation response in the northern subtropics is now more clearly visible between 1-100 hPa at all values of $\phi$ (and indeed the statistical significance of the anomaly in the subtropical upper stratosphere at $\phi = +60^{o}$ in December and January has increased). In November and December at $\phi=0^{o}$ the subtropical response at 30-40$^{o}$N is now weaker and more latitudinally confined (and in January it has completely disappeared), a demonstration that this subtropical response has been amplified
by processes that involve the polar vortex.

### 3.2 Tropospheric zonal wind response

At $\phi=0^{o}$ and $\phi=-60^{o}$ (figures 2b and 2c) there are highly statistically significant responses to the QBO that extend down into the troposphere, particularly in early and late winter. Figure 4 reproduces the regression results shown in Figure 2 but plotted
on a linear scale up to 30 hPa so that the tropospheric responses are highlighted (the colour scale has also been amended). The NH negative / positive dipole structure in November at 40$^{o}$N/ 60$^{o}$N shows a poleward progression as the QBO anomaly difference descends through the atmosphere from $\phi = +60^{o}$ down to $\phi = -60^{o}$ and concomitantly the subtropical jet in November is weakened at $\phi = +60^{o}$ and strengthened at $\phi = -60^{o}$. At both these $\phi$ values a connection appears between the tropospheric anomaly and the QBO anomaly in the very lowermost levels of the equatorial stratosphere, giving a horse-shoe
shaped response. This horse-shoe response at $\phi = -60^{o}$ is most clearly evident in late winter / spring e.g. in March. At the same time as the development of this horseshoe response an easterly anomaly develops in the upper equatorial tropical winds, reaching 3ms$^{-1}$ (99% statistical significance) in March at ~200 hPa. Inclusion of the additional polar vortex index (figure 5) reduces the significance of the subtropical jet strength anomaly, implying that there is some contribution from the polar vortex, but it nevertheless remains clearly present, especially in late winter (February, March at $\phi = -60^{o}$), and the
negative anomaly in the tropical upper troposphere remains largely unaffected. This suggests a more local response of the tropical troposphere and the NH subtropical jet in late winter, either in response to the presence of the overlying QBO anomaly in the lowermost stratosphere, for example through its impact on deep convection or in response to the QBO-induced meridional circulation (or both). It is not possible to determine from these diagnostics whether the timing of the maximum amplitude of this response in late winter has some underlying physical cause or whether it is simply an accident of
timing of the QBO descent to the lowermost stratosphere; carefully designed model experiments would be required to clarify this. Tropospheric variability is also reduced towards the end of winter, allowing the signal to be more easily detected.



An initial assessment of the horseshoe response at $\phi$ = -60$^o$ would suggest a straightforward interpretation in terms of a modulation of the Hadley Circulation strength by the QBO. In the stratosphere, the descending QBO-W phase gives rise to a local secondary meridional circulation anomaly in the tropics / subtropics with anomalous descent over the equator, below the level of the maximum westerlies, that suppresses the background upwelling in the equatorial lower stratosphere (and vice

versa under QBO-E conditions). QBO-W anomalies in the lower stratosphere also give rise to a weakened Brewer-Dobson (B-D) circulation as a result of reduced planetary wave forcing (White et al. 2015). Both impacts will thus weaken the upwelling branch of the B-D circulation. However, a simple interpretation in terms of this weakened B-D circulation response in the stratosphere somehow extending vertically down into the troposphere. Similarly, weakening the Hadley circulation cannot explain the response in figure 5c, since it indicates a strengthening of the Hadley Circulation.

This apparent contradiction has been recognised by previous authors and an explanation has been sought in terms of a QBO modulation of 'deep convection' i.e. the height to which equatorial tropical convective upwelling can extend. With the suppression of deep convection in QBO-W years due to the anomalous downwelling, the outflow into the subtropics that forms the upper horizontal branch of the Hadley circulation is strengthened, thus explaining the anomalously strong

subtropical jet in QBO-W years seen in figure 5c. Previous studies have indeed found that QBO-W years are less favourable for deep convection and have investigated mechanisms in terms of the QBO modulation of the zonal wind vertical shear, cold-point (tropopause) temperatures and associated changes in static stability (Nie and Sobel 2015; Yoo and Son 2016).

At $\phi$ = 0$^o$ in figure 4, there is some evidence of a similar horseshoe shaped response, but this is relatively weak (apart from in

March, where the pattern is similar to the $\phi$ = -60$^o$ response but weaker and less significant). Overall the QBO response at $\phi$ = 0$^o$ is dominated by the poleward (positive) part of the 40$^o$N/ 60$^o$N dipole response, which weakens from November through mid-winter, even though there is clearly a strong, statistically significant anomaly in the overlying stratospheric polar vortex in December / January. When the additional polar vortex index is included in the regression (figure 5b) the mid-winter (December / January) polar vortex anomaly in the stratosphere disappears, as expected, and so does the positive (but

statistically insignificant) anomaly immediately below it in the troposphere. In February (figure 4b), the ~60$^o$N anomaly at $\phi$ = 0$^o$ is negative (but is not statistically significant) even though the overlying anomaly in the stratosphere is positive. This does not tally particularly well with the mechanism for QBO influence via the polar route that involves a QBO influence on the polar vortex strength (usually via a change in frequency of SSWs and their downward influence into the troposphere). Inclusion of the NH vortex index underscores this, since the amplitude and statistical significance of this mid-latitude

response in the lower troposphere in February and March (figure 5b) remains relatively unchanged.

The 40$^o$ / 60$^o$ dipolar response at $\phi$ = 0$^o$ in the NH troposphere in November (figure 4b) is noticeably weakened when the polar vortex index is introduced (figure 5), confirming that a substantial part of this early winter dipolar response is connected to the stratospheric vortex variability but a 95% statistically significant (positive) response nevertheless remains at





around 50°N in November at both $\phi = 0°$ and $\phi = +60°$. The high latitude positive anomaly at $\phi = +60°$ has a mirror-image negative anomaly at $\phi = -60°$ so it is not clear whether the anomaly is a result of a sensitivity to the QBO in the upper stratosphere or the very lowermost stratosphere. Nevertheless, the analysis shows a clear QBO anomaly at high latitudes that is not directly associated with variations in polar vortex strength at 60°N, 10 hPa. This remaining high latitude signal could

be due to the choice of an imperfect vortex index that does not characterise the vortex behaviour sufficiently well, or it could be due to the presence of additional mechanisms that are active in early winter, such as planetary wave reflection from the upper stratosphere (Perlwitz et al. 2003;2004; Shaw et al. 2014; Lu et al. 2017) that do not involve the mediation of large-scale variations of the polar vortex such as SSWs.

While the focus of this study is on the Northern Hemisphere, we note the presence of a similar late winter (December) dipolar anomaly in the Southern Hemisphere (SH) at $\phi=-60°$ centred around the SH mid-latitude jet that is similarly unaffected by inclusion of a SH polar vortex index in the regression (not shown).

### 3.3 Northern Hemisphere sea level pressure response

Figure 6 shows results for November - March from the regression analyses of mean sea level pressure over the NH with (a)
$\phi=+60°$, (b) $\phi=0°$ and (c) $\phi=-60°$. The previously identified surface response in January over the Atlantic / European sector (e.g. Holton and Tan 1982; Anstey and Shepherd 2014 – see their figure 2) is evident at $\phi=0°$ and is statistically significant at the 95% level (fig 76). The response resembles a positive NAO response to QBO-W conditions, with anomalously low pressure over the pole and a strengthening of the subtropical high over the Azores (and vice versa under QBO-E conditions). The timing and optimum $\phi$ value of this response corresponds well with the timing of the maximum response of the
stratospheric polar vortex. There is also a positive response over China and the west Pacific, giving the suggestion of an annular mode response (and this is also evident at $\phi=+60°$ in January and February) but neither response is statistically significant.

Figure 7 shows the corresponding QBO signal when the additional NH vortex index was included. Note that we have not
imposed a lag on the polar index to take into account the delay between variability in the polar vortex and the troposphere / surface. A lagged response would be characterised by a shift in $\phi$ of the maximum response. (Tests with imposed lags showed very little sensitivity, possibly because the data are monthly averaged and the averaging is therefore on the same timescale as the lag in the response).

The positive NAO-like response in the QBO regression coefficients in January at $\phi=0°$ is substantially weakened by the inclusion of the polar vortex index and it is no longer statistically significant, demonstrating that the QBO variability in the stratospheric polar vortex plays an important role in this January NAO-like response in mslp seen in figure 6b.




At $\phi = +60^o$ (figures 7a) there is also a mslp response pattern in December, not previously reported. It resembles a positive annular mode with substantial response amplitudes (>5 hPa difference between QBO-W and QBO-E) in both the North Pacific and the Atlantic / European sector. The statistical significance of the December response over the Atlantic / European sector is greater than 99% i.e. it has a higher statistical significance than the previously identified NAO-like

response in January, described above. However, the response is centred further eastward over Europe, so that it likely projects onto the East Atlantic pattern rather than the NAO itself, and it also likely projects onto the major centres of blocking events in these two regions.

Referring back to figure 1, the $\phi=+60^o$ QBO vertical wind profile is typical of a descending westerly phase in the upper

stratosphere i.e. at an earlier point in time than the $\phi = 0^o$ profile. However, this sensitivity of the December response to higher levels in the equatorial stratosphere than the January response is unlikely to be simply due to the gradual descent of the QBO winds from December to January in the same year. As discussed in section 2, a change of $\phi = 10^o$ is equivalent to a time shift of ~0.8 months, so the time difference between $\phi = +60^o$ and $\phi = 0^o$ is at least 4 months.

A possible reason that this December mslp response has not previously been identified is as follows. $\phi = +60^o$ characterises the descending phase progression of the QBO in which the west-east amplitude differences are maximum in the upper stratosphere between ~15-30 hPa. In the lower stratosphere the corresponding anomalies are weak easterlies or close to zero (see figure 1). Thus a composite analysis based on the sign of the lower stratospheric QBO winds (at, say, 50hPa) is likely to miss (or substantially underestimate) this December mslp response since winters with this height profile are likely to be

excluded by their small wind values in the lower stratosphere. This is confirmed by the response at $\phi = 0^o$, which shows no clearly discernable response to the QBO in December.

In the regression analysis in which the NH polar vortex index was included, the $\phi=+60^o$ response in December (figure 7a) remains virtually unchanged. This suggests that it is not a direct result of the influence of vortex variability extending down

to the surface, as was the case for the January $\phi=0^o$ response. This early winter mslp sensitivity to QBO winds in the upper stratosphere rather than the mid-to-lower stratosphere shows similarity to the results of Lu et al. (2017), who examined the mslp response to the 11-year solar cycle in terms of downward wave reflection. In particular, they found a dual response over the Aleutian Low region in the North Pacific and over the Atlantic / European sector in early winter, similar to figure 6a in December (see their figures 3 and 4). They also noted that the mslp anomaly pattern differs from the classical NAO

pattern because it extends further eastward towards Eastern Europe, again in a similar fashion to figure 6a in December.

While circumstantial, this similarity in mslp response suggests that a similar planetary wave reflection mechanism could be responsible for the highly significant QBO response seen at $\phi = +60^o$ in December, a possibility that deserves further investigation. However, inclusion of a simple index in the regression analysis to represent years that are favourable for




downward wave coupling, as proposed by Perlwitz and Harnik (2003; namely the difference of the zonal wind at 2 hPa and 10 hPa averaged between $58^o$-$74^o$N), did not produce any notable changes to this December response (we would expect it to weaken substantially, in the same way as the NAO response in January weakens when the 10 hPa, $60^o$N vortex index was included). Nevertheless, this simple 2-10 hPa zonal wind index may not adequately represent downward wave coupling.

Shaw et al. (2010) suggested that it may require a more complex index that includes information about the vertical and meridional wavenumbers of the relevant waves.

On the other hand, there is also a similar response of the opposite sign over Europe at $\phi$ = -$60^o$, the mirror image of the $\phi$ = +$60^o$ response, so it is not clear whether this December response indicates a sensitivity to the upper stratosphere ($\phi$ = +$60^o$)

or to the very lowermost stratosphere ($\phi$ = -$60^o$). Additionally, the solar signal of Lu et al. (2017) appears to be associated with downward wave reflection and associated wave divergence response in the polar lower stratosphere; given that there is no significant change in the extratropical stratospheric zonal winds during this time (see figure 2a) and the mslp signals are located near the major centres of blocking, we suspect that nonlinear processes such as baroclinic and/or barotropic instability in the high-latitude upper troposphere and lower stratosphere may be responsible, but more research is needed to

verify this.

Later in the winter season (February / March) a statistically significant negative anomaly is evident at $\phi$ =-$60^o$ (figure 6c) over the Aleutian Low region of the North Pacific. The $\phi$ =-$60^o$ QBO is typical of the later stages of a descending QBO-W phase, with peak QBO-W minus QBO-E differences in the height region ~70 hPa. The mslp anomaly remains unchanged

when the polar vortex index is included (figure 7c), indicating that it is unlikely to be directly related to the variability of the polar vortex. This region of the Pacific is highly coupled to tropical variability, largely through the presence of Rossby wave-trains generated by anomalous tropical heating, so the continued presence of the anomaly despite inclusion of the vortex index, together with its association with the QBO winds in the very lowermost region of the equatorial stratosphere, suggests that its origin is in the tropics or subtropics. This is further confirmed if the regression analysis is repeated with an

additional index to represent the strength of the subtropical jet in the Pacific sector (using the 200 hPa zonal winds at $30^o$N averaged between $160^o$E – $160^o$W). In this case, the Aleutian Low response at $\phi$ = -$60^o$ in March was greatly reduced in amplitude and lost its statistical significance (not shown). We note that it was not sufficient to use a zonally-averaged NH subtropical jet index; the removal of the Aleutian Low response in this way required an index based only on the Pacific Ocean subtropical jet, confirming the results of Garfinkel et al. (2012) who showed that the response of the subtropical jet to

the QBO differed between the Pacific and Atlantic basins.

As a check on these mslp results and to help with their interpretation, Figure S5 shows results from regression analyses in which the QBO index was derived in the more traditional manner, using the $U_{eq}$ time-series from a single pressure level (20hPa, 30hPa, 40hPa, 50hPa and 70hPa respectively) instead of combining them using the EOF approach. The results are





entirely consistent with the EOF analysis. The response in early winter (Dec) is captured particularly well by the $U_{eq}$ = 20hPa time-series, the mid-winter response (January) is maximised at 40hPa and the negative Pacific response in late winter (February / March) is captured by the $U_{eq}$ at both 50hPa and 70hPa. The NAO-like response is evident in January throughout the whole range 20-50 hPa (although it is only statistically significant at 40 hPa and 50 hPa). This is also consistent with the

results described above, since fig 1 shows that the vertical profile of ueq at $\phi=0^o$ has a relatively deep westerly anomaly throughout the range 15-70 hPa. This suggests that the mslp response in mid-winter is particularly sensitive to the presence of a vertical QBO wind profile that has the same sign anomaly throughout the depth of the stratosphere up to at least 10 hPa, thus explaining why the January results in figure S5 all have similar patterns of the same sign and only differ in amplitude. In contrast, both early winter (December) and late winter (February / March) responses show a $180^o$ phase difference

between the 20 hPa and 70 hPa responses. This emphasizes, as discussed above, that these early / late winter responses are unlikely to result from the same mechanism as the January response.

### 3.3 Convective precipitation response

Regression analysis results for precipitation are more uncertain, given the difficulty of accurately representing this quantity in the reanalysis models because of their reliance on parameterisation of local processes that are too small to capture

explicitly. Nevertheless, the results provide QBO response patterns that are generally consistent with previous studies of satellite observations, such as those of Leiss and Geller (2012) who performed an annual-mean composite analysis of the GPCP observations (Global Precipitation Climatology Project; Huffman et al. 1997) and Seo et al. (2013) who examined the QBO impact on rainfall in the western North Pacific in spring using the same dataset. Figure 8 shows the QBO response in convective precipitation for all months using $\phi = -60^o$ i.e. the QBO index most highly correlated with the equatorial winds in

the lowermost part of the stratosphere at ~70 hPa. Although there is little statistical significance in any of the individual months, the response patterns between March and September are generally consistent and show increased tropical rainfall over Indonesia, the West Pacific and Central America under QBO-W conditions compared to QBO-E conditions, particularly during boreal summertime. Figure 9 shows the corresponding QBO response in total (i.e. convective plus large-scale) precipitation; perhaps unsurprisingly, the response, while noisier, is similar in pattern to figure 8. Comparison of

figures 8 and 9 shows that the major contribution to the QBO response is from the convective precipitation but the large-scale rainfall response contributes, in particular, to the negative anomaly that stretches east-west across the Pacific at ~$10^o$N which is more prominent in the total precipitation response.

The response in figure 8 has the opposite sign to that found by Leiss and Geller (2012; see their figure 7) and to Seo et al.

(2013). This could be explained by the different QBO indices employed. For example, in contrast to our EOF approach, Seo et al. (2013) employ $U_{eq}$ at 50 hPa to characterise the QBO while Leiss and Geller used the $U_{eq}$ at 70 hPa lagged by 3 months in order to account for the downward propagation of the QBO signal down to the tropopause. We have not employed a time-lag in our study, since in theory the lagged response should be identified by a shift in $\phi$ of the maximum response. However,





the maximum response is generally seen over a rather broad range of ϕ so identification of a lagged response in this way is rather difficult. Additionally, the lowest level of the equatorial winds used to calculate the EOFs (see section 2) was 70 hPa, since equatorial winds below this level are much more variable and characterisation of the QBO using only two EOFs was found to be less reliable.

In contrast, figure 10 shows the QBO response in convective precipitation when a QBO index at ϕ = +30 is employed i.e. more characteristic of the equatorial winds at 30-40 hPa. A response of the same sign as that of Leiss and Geller (2012) is evident. Reduced rainfall under QBO-W conditions is seen over Indonesia, the West Pacific and Central America between June – November, although the statistical significance is minimal over these regions. In addition, a marked shift in the ITCZ
is also present and is highly significant between August and November suggesting a southward shift of the upwelling branch of the Hadley Circulation under QBO-W conditions compared with QBO-E conditions. While outside the scope of the current study, an understanding of these QBO responses in tropical precipitation requires further investigation, for example through more detailed comparisons with the GPCP observations and study of the associated zonal and meridional circulation anomalies.

**4 Summary**

QBO teleconnections with the NH stratospheric polar vortex and the Earth's surface have been explored using regression analyses of the stratospheric polar vortex zonal winds, tropospheric zonal winds, mean sea level pressure and precipitation.

Using an EOF approach, a single QBO index was derived to represent the QBO, employing $u^* = r \sin(\psi + \phi)$ where $r^2 = P_1^2$
$+ P_2^2$, $P_1$ and $P_2$ are the principal components of the first two EOFs of the equatorial wind in the region 10-70 hPa, $\psi = \arctan$ $(P_2/P_1)$ and ϕ is an arbitrary phase shift that can be interpreted as a rotation in phase space (see section 2 for further details). By performing repeated regression analyses, the optimum ϕ to maximise the observed response can be determined. Using this combined EOF time-series as a QBO index instead of the usual equatorial wind at a single level allows the vertical height structure of the equatorial winds to be taken into account. Sample results at ϕ= +60°, ϕ = 0° and ϕ = -60° are compared
and contrasted as the primary diagnostic, and these can be broadly equated to using a QBO time-series index $U_{eq}$ at 20 hPa, 40 hPa and 70 hPa respectively.

The optimum value of ϕ for the polar vortex QBO response was found to be ϕ = 0° (Figure 2), corresponding to a vertical wind profile $U_{eq}$ with a maximum QBO anomaly in the mid- to lower stratosphere (~40 hPa). Westerly QBO anomalies are
associated with a relatively strong, undisturbed polar vortex (see figure 1), in good agreement with the results of Holton and Tan (1980; 1982). These results from the analysis of 58 years of data were remarkably consistent with the results of Baldwin and Dunkerton (1998) who only analysed18 years.



A QBO response in the zonal wind field was also seen extending deep into the troposphere at subtropical, mid- and high-latitudes, particularly in early winter (November) at $\phi = 0^o$ and $\phi = -60^o$ and in late winter (March) at $\phi = -60^o$ (figure 4). As the QBO descends through to the lowermost levels of the stratosphere, a horseshoe shaped response is seen to emerge,

connecting the lowermost stratospheric QBO anomaly with an anomaly in the subtropical jet strength that extends deep into the troposphere. An easterly wind anomaly also emerges in the tropical upper tropospheric winds at $\phi = -60^o$ and this strengthens during the winter, reaching a maximum of $\sim 3$ ms$^{-1}$ in late winter (March) with statistical significance greater than 99%.

In addition, a dipolar response at 40$^o$N / 60$^o$N was evident, and was strongest and most significant at $\phi = 0^o$ i.e. at the same value of $\phi$ for which the maximum stratospheric polar vortex response was observed. This dipolar response was particularly evident in early winter (November), but declined in amplitude (and statistical significance) throughout mid-winter, even though a clear stratospheric polar vortex anomaly was evident aloft.

A number of different mechanisms for these tropospheric signals have been proposed in previous studies. These include (a) the downward influence of the stratospheric polar vortex anomaly that arises from the Holton-Tan mechanism discussed above (the polar route), (b) an influence from the QBO-induced meridional circulation in the subtropics that influences the subtropical jet (the subtropical route), and (c) an influence of the QBO on deep convection which can also influence the sub-tropical jet (the equatorial route).

A novel approach was employed in this study in an attempt to eliminate the major impact of one or more of these mechanisms, with an emphasis on eliminating the impacts of the polar route. This was done by introducing an additional index in the regression analysis to characterise inter-annual variations of the NH polar vortex, using an index of the zonally averaged vortex wind at 10 hPa, 60N. By including this index, any QBO influence on the troposphere that arises primarily

via a modulation of the polar vortex (often through the occurrence of SSWs) was effectively removed, since this variability matched the polar vortex index more closely than the QBO index. The horseshoe response at $\phi = -60^o$ (figure 4c, March) and the upper tropospheric tropical wind anomaly were largely unaffected by inclusion of this vortex index (figure 5), demonstrating that this QBO influence is likely to be via the tropical or subtropical routes rather than the polar route.

When the additional polar vortex response was introduced at $\phi = 0^o$, the 40$^o$N / 60$^o$N dipolar response was eliminated, demonstrating that this response is primarily associated with polar vortex variability. On the other hand, the early winter response at $\phi = +60^o$ remained largely unaffected (e.g. in December, figure 5a). This suggests that either (a) the additional polar vortex index may not have completely captured (and removed) the tropospheric signal associated with a QBO modulation of the stratospheric polar vortex, or else (b) there is an additional mechanism in early winter whereby the QBO


influences the troposphere, such as planetary wave reflection from the upper stratosphere, that are not reliant on substantial variations in the polar vortex. However, interpretation of this response is complicated by the fact that the $\phi = -60^o$ and $\phi = +60^o$ responses are similar but of opposite sign (particularly in December) so it is impossible to determine whether the observed responses indicate sensitivity to the QBO in the upper stratosphere ($\phi = +60^o$) or to the lowermost stratosphere ($\phi = +60^o$); model experiments would be required to resolve this, such as those performed with a mechanistic model by Gray et al. (2004).

At the surface, several QBO signals in the mean sea level pressure were identified. At $\phi = 0^o$ the previously-reported positive NAO response in January associated with a westerly QBO phase in the lower stratosphere was observed (figure 6b). This NAO-like response was evident at all values of $\phi$ in the range $-60^o < \phi < +60^o$, unlike the responses in early and late winter, when there was a $180^o$ phase change between $\phi = -60^o$ and $\phi = +60^o$. Examination of the typical vertical profiles of the QBO wind anomalies at these values of $\phi$ (figure 1) shows that the positive NAO-like response in January responds to the presence of westerly QBO anomalies through a relatively deep height range, including 5-50 hPa ($\phi = +60^o$), 10-70 hPa ($\phi = 0^o$) and 30-70 hPa ($\phi = -60^o$), and maximises when there are deep westerlies throughout this height range ($\phi = 0^o$). As already noted

in earlier studies (e.g. Gray et al. 2004) this result is perhaps unsurprising since the vertically propagating planetary-scale waves involved in the 'polar route' of influence are known to have deep vertical structures.

In addition, two further responses were found, which have not previously been reported. In December a statistically significant, positive west minus east anomaly difference was found in mslp fields over both Europe and the North Pacific

(figure 6a) at $\phi = +60^o$. The anomaly is thus maximised with a QBO index that correlates well with the equatorial winds at 20 hPa, a level much higher than normally employed to characterise the QBO. The response over the Atlantic / European sector is shifted eastward compared with the NAO pattern. Neither the European nor Pacific responses were affected by inclusion of the stratospheric polar vortex index. This could be because the polar vortex index has not completely captured all variability associated with the polar vortex, or there is a further mechanism that may be operating in early winter, possibly

affecting both tropospheric winds and mean sea level pressure. Note that the high latitude tropospheric wind anomaly at $\phi = +60^o$ in December was similarly unaffected by inclusion of the polar vortex index (figure 5) and these two responses may therefore be related. Subtle changes in the region with small values of potential vorticity gradients in the middle to lower stratosphere is one possible influence that deserves further investigation in this regard.

In March, a negative QBO-W minus QBO-E mslp anomaly difference is also seen over the North Pacific (at $\phi = -60^o$; figure 6). Using the same technique of introducing an additional index to remove, in this case, variability associated with the Pacific subtropical jet, this anomaly was shown to be closely associated with a strengthening of the Pacific subtropical jet. We note the anomaly was only eliminated when using a Pacific subtropical jet index and not a zonally-averaged index, i.e.





the QBO response of the subtropical jet in the Pacific and the Atlantic differ. This emphasises the importance of processes local to the Pacific; a strong candidate is the influence of the QBO on deep convection in the tropical Pacific.

Finally, the QBO impact on precipitation was examined, albeit with awareness of the difficulty of representing these in
reanalysis models because of the challenges of parameterising processes that occur at sub-grid scales. An increase in tropical convective and total rainfall was found under QBO-W conditions, over Indonesia, the tropical Pacific and Central America, from March through to September (figure 8). This response is in the opposite sense to that found by Leiss and Geller (2012); the response was found to maximise at $\phi = -60^o$ i.e. at ~70 hPa, whereas Leiss and Geller used a QBO index based on 70 hPa with an additional 3 month lag to represent the downward extension of the QBO to the tropopause. The use of this time lag
could be a partial explanation for the difference in our results, but may also be a result of the different approach to specifying the QBO, since we employ an EOF-based index that can capture characteristics of the vertical profile of the QBO winds more easily than using an index from a single level.

In addition, a statistically significant southward shift of the ITCZ was found under QBO-W conditions compared to QBO-E
conditions, particularly between August – November. This response maximised at $\phi = +30^o$ i.e. at levels most closely correlated with QBO winds at 30-40 hPa. Inclusion of an additional NH polar vortex index did not affect it, thus demonstrating that the response is unlikely to be associated with a modulation of the large-scale Brewer-Dobson Circulation forcing. These QBO responses in precipitation warrant further investigation, for example by carefully comparison with observations and analysis of corresponding anomalies in both zonal and meridional circulation anomalies.

**Acknowledgments**

LG, HL and SO acknowledge funding from the UK Natural Environment Research Council (NERC) through support of the ACSIS (Atlantic Climate System Integrated Study) Programmes at the National Centre for Atmospheric Science (LG), the British Antarctic Survey (HL), and the Belmont Grant NE/P006779/1 – GOTHAM – 1505DC004/MW2 (SO). YK was supported by JSPS KAKENHI Grant Numbers 26287117 and 15KK0178 and by the Environment Research and Technology
Development Fund (2-1503) of Environmental Restoration and Conservation Agency. VS acknowledges doctoral study support at Oxford through a Rhodes Scholarship.

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

5  **distance from the origin represents the amplitude r (see text). Representative vertical profiles are provided to show the typical vertical wind profile for that value of ϕ. The pressure levels shown on each vertical profile panel are 70, 50, 40, 30, 20, 15 and 10 hPa.**





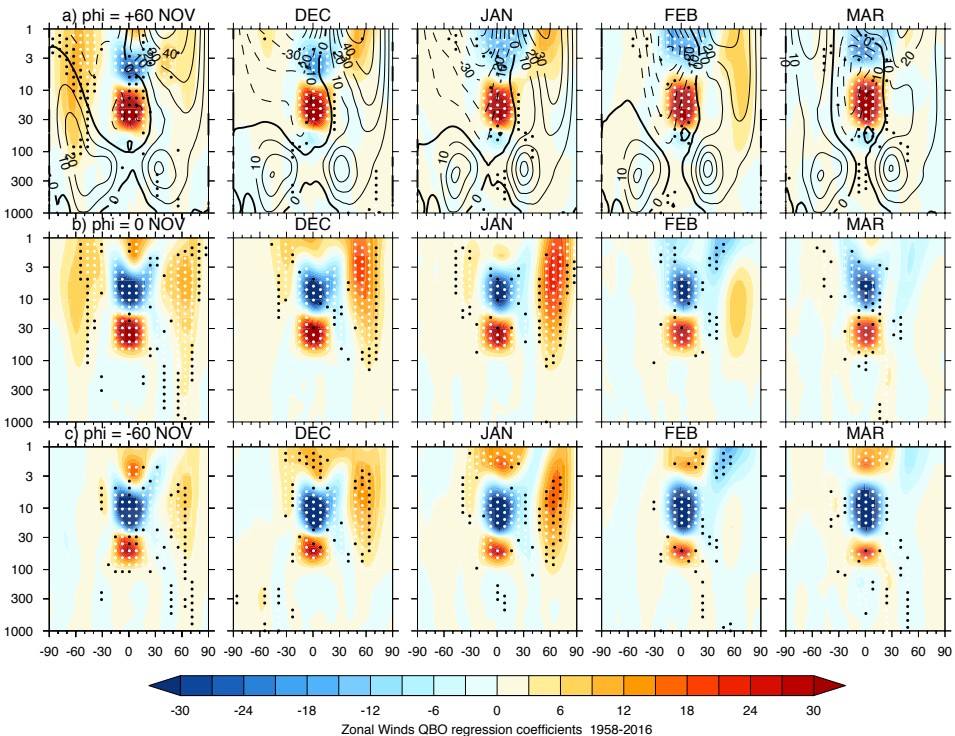

**Figure 2: Regression based QBO-W minus QBO-E differences: height – latitude cross-sections of zonally-averaged zonal winds (ms⁻¹) for November – March from the period 1958-2016. Rows (a) – (c) show results using φ = +60°, φ = 0° and φ = -60° respectively in the definition of the QBO index (see text). Values of φ = +60°, 0° and -60° are broadly equivalent to using $U_{eq}$ = 20 hPa, 40 hPa and 70 hPa as the QBO index, respectively. Contours in panel (a) show the climatological values. White (black) dots indicate 99% (95%) statistical significance.**





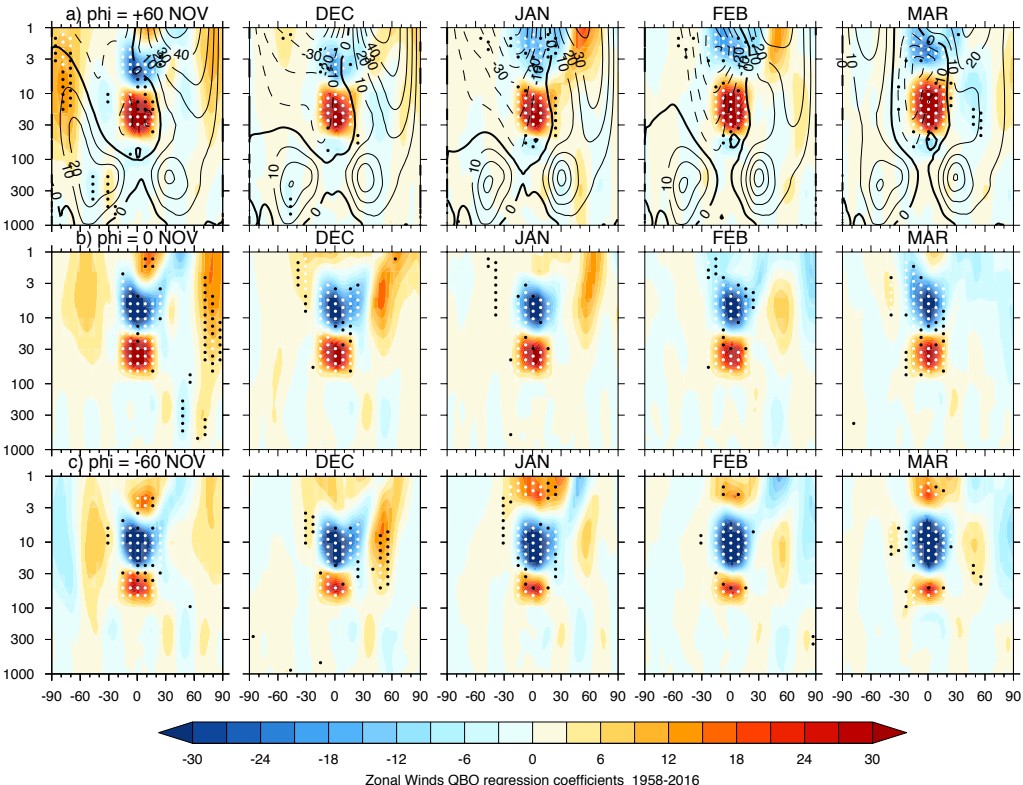

**Figure 3: Regression based QBO-W minus QBO-E differences when an additional index is included to represent the polar vortex variations: height – latitude cross-sections of zonally-averaged zonal winds (ms$^{-1}$) for November – March from the period 1958-2016. Rows (a) – (c) show results using $\phi$ = +60º, $\phi$ = 0º and $\phi$ = -60º respectively in the definition of the QBO index (see text). Values of $\phi$ = +60º, 0º and -60º are broadly equivalent to using $U_{eq}$ = 20 hPa, 40 hPa and 70 hPa as the QBO index, respectively. Contours in panel (a) show the climatological values. White (black) dots indicate 99% (95%) statistical significance.**





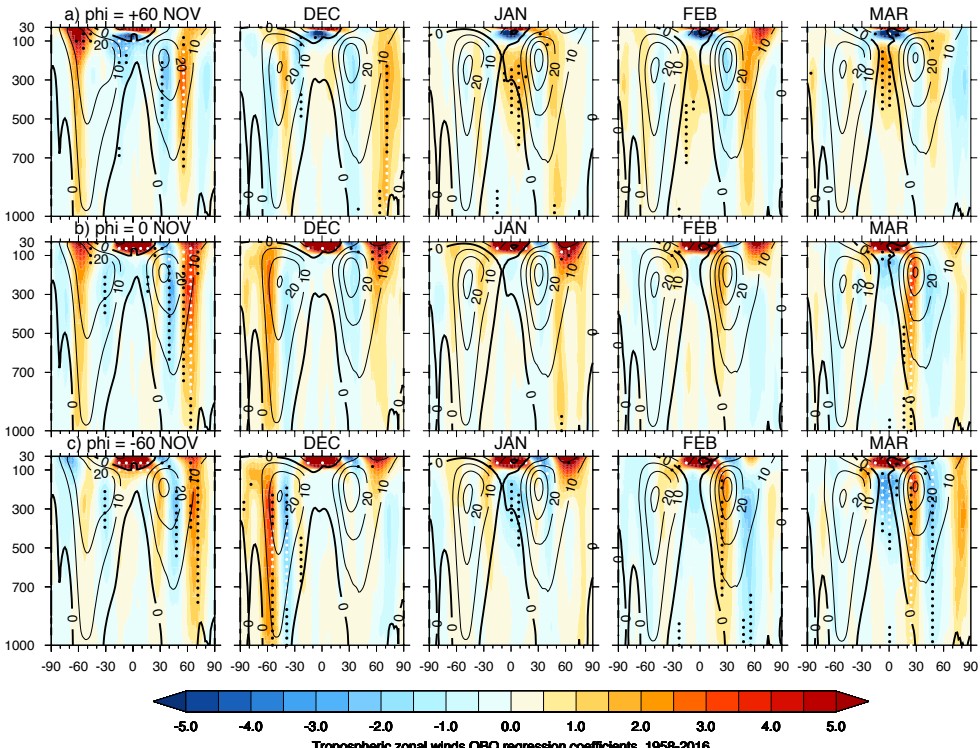

**Figure 4: As figure 2 but using a linear height scale between 30-1000 hPa to emphasise the tropospheric response. Note the colour scale has been amended accordingly.**



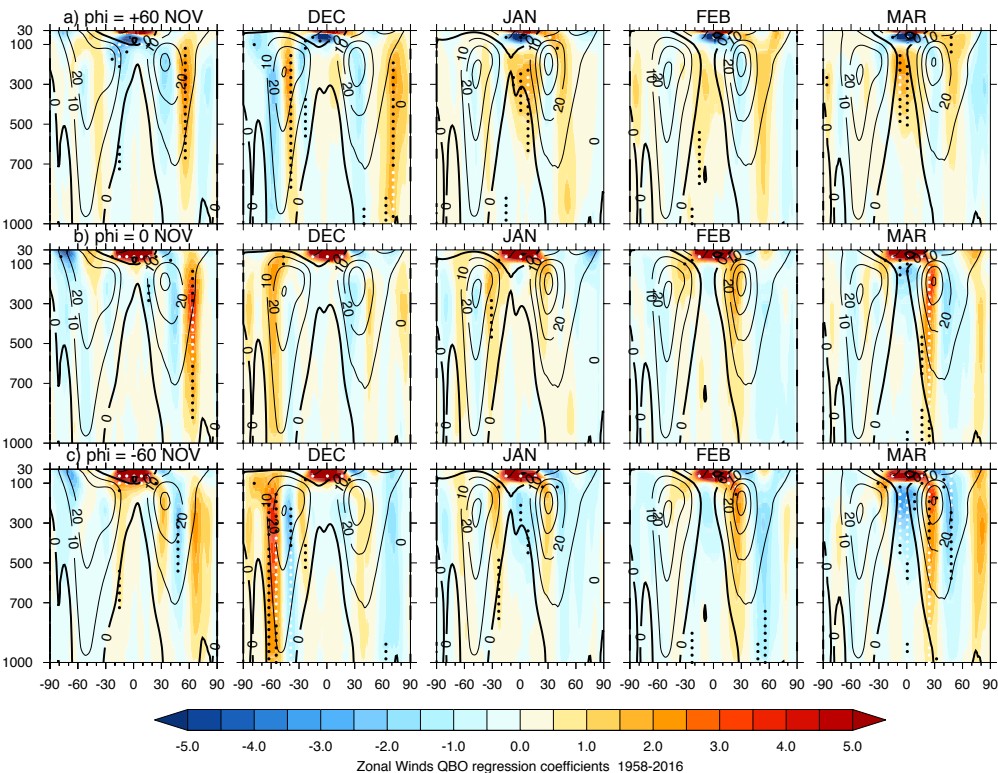

**Figure 5:** As figure 3 but using a linear height scale between 30-1000 hPa to emphasise the tropospheric response. Note the colour scale has been amended accordingly.





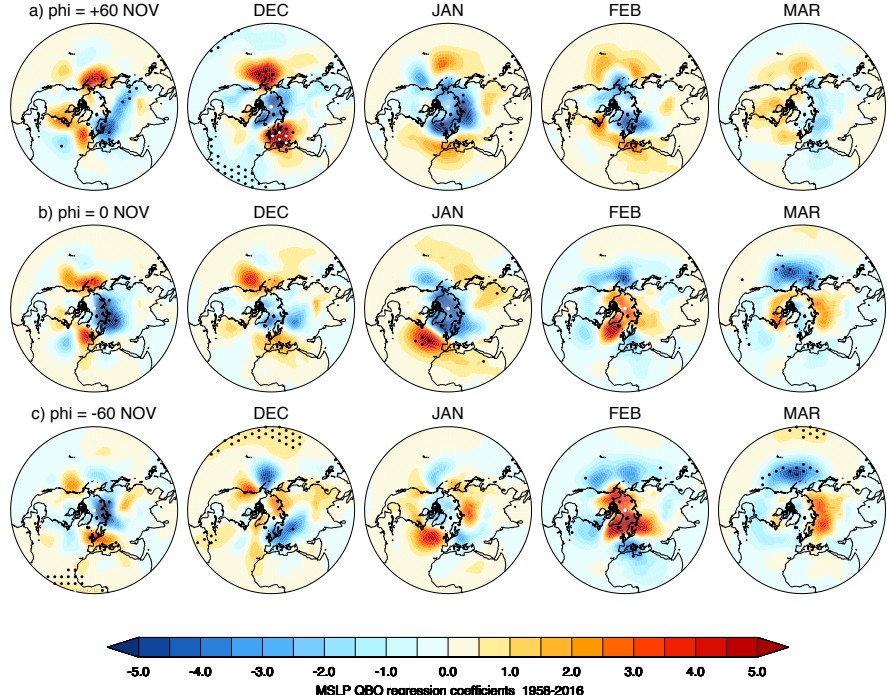

**Figure 6: Regression based QBO-W minus QBO-E differences: latitude-longitude distribution of mean sea level pressure (hPa) for**
5  **November – March from the period 1958-2016. Rows (a) – (c) show results using ϕ = +60°, ϕ = 0° and ϕ = -60° respectively in the definition of the QBO index (see text).  Values of ϕ = +60°, 0° and -60° are broadly equivalent to using $U_{eq}$ = 20 hPa, 40 hPa and 70 hPa as the QBO index, respectively. White (black) dots indicate 99% (95%) statistical significance.**



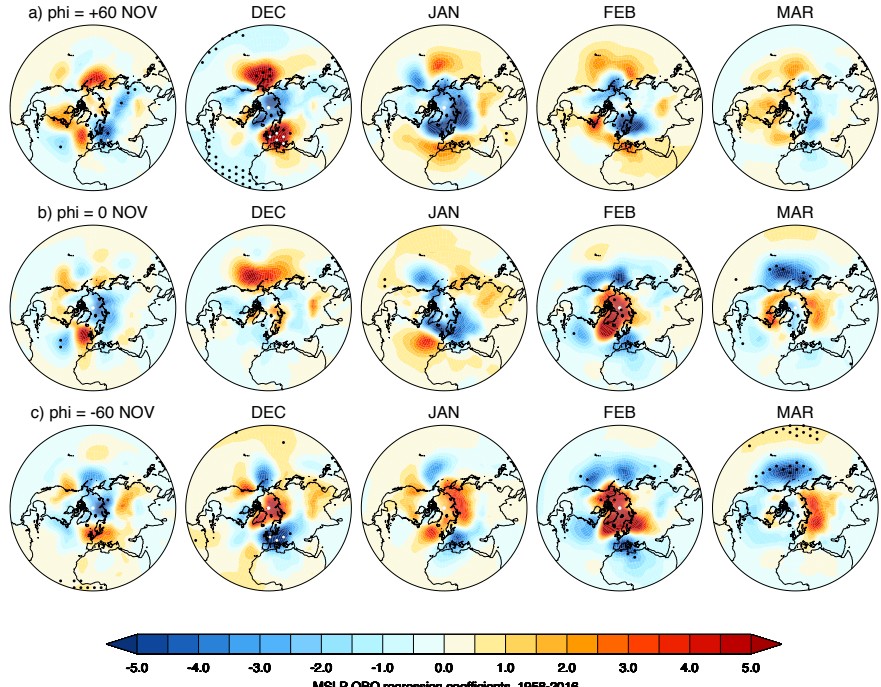

**Figure 7: Regression based QBO-W minus QBO-E differences when an additional index is included to represent the polar vortex variations: latitude-longitude distribution of mean sea level pressure (hPa) for November – March from the period 1958-2016. Rows (a) – (c) show results using ϕ = +60º, ϕ = 0º and ϕ = -60º respectively in the definition of the QBO index (see text). Values of ϕ = +60º, 0º and -60º are broadly equivalent to using $U_{eq}$ = 20 hPa, 40 hPa and 70 hPa as the QBO index, respectively. White (black) dots indicate 99% (95%) statistical significance.**





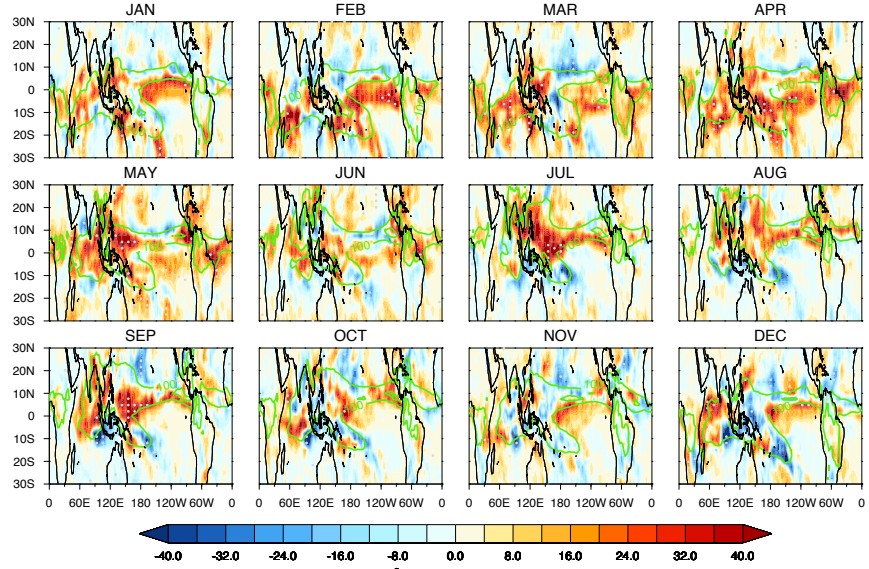

**Figure 8: Regression based QBO-W minus QBO-E differences: latitude-longitude distributions of convective precipitation (mm per month) for all months using $\phi = -60°$ in the definition of the QBO index. This is broadly equivalent to using $U_{eq} = 70$ hPa as the QBO index. White (black) dots indicate 99% (95%) statistical significance. Convective precipitation fields were from the ERA-Interim dataset for the period 1979-2016. Black contours show land outlines; green contours show climatological values with contour interval of 100 mm per month.**



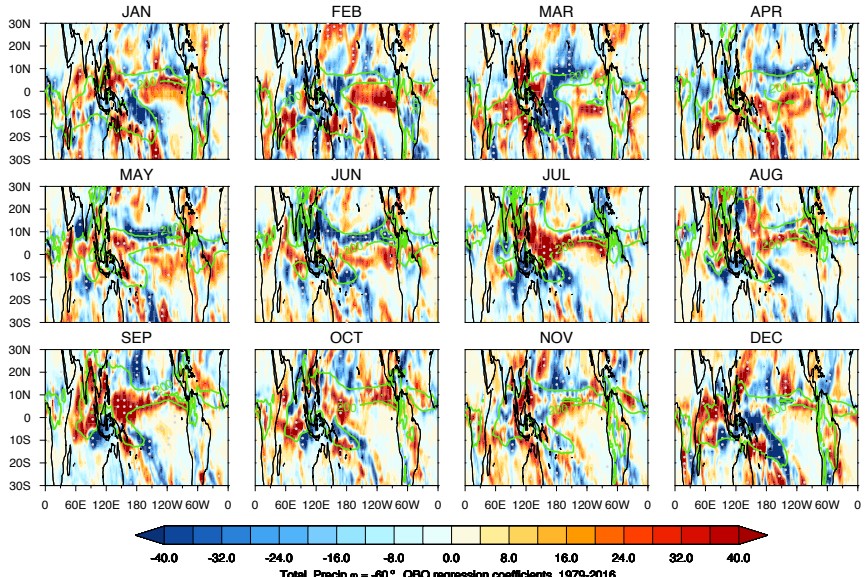

**Figure 9: As Figure 8 but for total (i.e. convective plus large-scale) precipitation. Black contours show land outlines; green contours show climatological values with contour interval of 200 mm per month.**




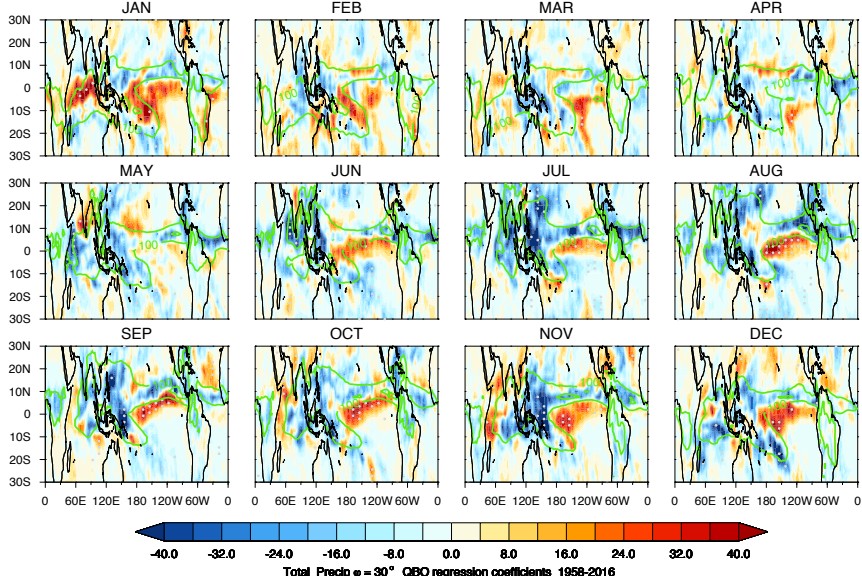

Figure 10: as Figure 8 but using $\phi$ = +30° in the definition of the QBO index. This is broadly equivalent to using $U_{eq}$ = 20 hPa as the QBO index.