# Peer review of "Surface impacts of the Quasi Biennial Oscillation"

_Atmospheric Chemistry and Physics, 2017_

## Referee Comment (RC1) · Anonymous Referee #1 · 7 Jan 2018

Review of "Surface Impacts of the Quasi Biennial Oscillation," By Lesley J. Gray,
James A. Anstey, Yoshio Kawatani, Hua Lu, Scott Osprey, and Verena Schenzinger

This is a very interesting, and valuable, paper. While many previous papers
have looked at the influence of the stratospheric quasi-biennial oscillation (QBO) on
various aspects of atmospheric behavior, such as on stratospheric jet structure,
surface weather, and tropical precipitation, this paper examines all of these in a
holistic manner. Furthermore, it seeks to shed some light on the possible
mechanisms whereby the QBO may exert its influence. It does this by using
multivariate linear regression analysis of the QBO with the analyzed fields. It
includes other influences in this regression analysis, such as stratospheric volcanic
aerosol abundance, ENSO, solar activity, and a long-term trend. Other interesting
aspects to their analysis is that, rather than examining time series of the influence of
the QBO winds at individual levels, they utilize time series of the first two QBO
empirical orthogonal functions (EOFs) between 70 and 10 hPa. Noting that EOF1
maximizes in the upper part of this height range, while EOF2 maximizes in the lower
portion, they make conclusions about which height range of the QBO influences
various atmospheric fields.

There are a number of different pathways by which the QBO may influence
atmospheric behavior. The QBO variation in equatorial winds modulates the winter
waveguide for extratropical planetary waves, which can affect the strength of the
polar vortex and stratospheric polar temperatures. The authors call this the "polar
route." The QBO modulation of stratospheric equatorial winds is accompanied by
meridional circulation, as is required by the thermal wind relationship. This, in
turn, modulates equatorial stratospheric temperatures all the way down to the
region below the tropical tropopause. This modulates tropical rainfall and tropical
deep convection. This has been hypothesized in various observational analysis
papers, and has also been shown in cloud-resolving modeling studies. The authors
call this the "tropical route." The QBO equatorial meridional circulation has
equatorial upwelling that decreases tropical temperatures in easterly shear
conditions, while the return circulation increases temperatures through
downwelling in the subtropics. This affects the equatorial-subtropical temperature
gradient, which in turn affects the wind shears in that region, which affects
baroclinic waves and planetary waves. The authors call this the "subtropical route."

As mentioned previously, many previous authors have looked at QBO
influences on the atmospheric circulation, but this paper differs from these in
various ways. One is in the length of the data record being analyzed. The zonal
wind a fields they analyze is the combined ERA-40 and ERA-Interim analyses that,
together, extend from 1958-2016, while the precipitation is only the ERA-Interim
analysis (1979-2016). The mean sea level pressure fields are from the Hadley
Center for the period 1958 onward.

One interesting feature of their analysis is when they include both QBO and
polar vortex indices in their regression analyses. This serves to separate effects

where the QBO influence is through the "polar route," since QBO influences that remain after including the polar vortex index as a separate term in the regression analysis, would not likely involve the "polar route."

One shortcoming of this paper is its identification of QBO influences on so may aspects of the atmospheric circulation, that it is difficult to recall all of them. I suggest a summary table that includes all of the identified variables they identify as being influenced by the QBO along with their conclusions about the height range of the QBO that they identify as being key and also which of the three routes they believe to be most likely responsible.

These are my global comments. In the following, I make my more specific comments.

Specific Comments

1. Page 1, Lines 27-29: Given the fact that both QBO and solar indices are used in the authors' regression, I find it odd that Labitzke's work is not mentioned here. The Holton Tan mechanism seems to be opposite, depending on the phase of the solar cycle. This is particularly so since several of the authors have written on this subject. Also, Labitzke et al. (2006) is in the reference list, but I can't find it in the text.

2. Page 3, line 29: It seems to me that the reason why previous authors focus on the QBO at 40-50 hPa is that is where the amplitude of the QBO is maximum. That should probably be mentioned.

3. Page 4, line 32: A reference is needed for this statement.

4. Page 5, line 4: It's interesting that 615 stations have provided data for more than 100 years. It doesn't seem that data for all those years are used in this paper though.

5. Page 5, line 24: I think that the statement "that these indices are independent of one another" is too strong. Certainly, this is contradicted by statements in Garfinkel and Hartmann (2007), Salby (1996), and Taguchi (2010). It is sufficient to say that the inclusion/exclusion tests were done.

6. Page 6, lines 4-11: Both EOFs have opposite signs at different altitudes. This is contrary to the implication that only EOF1 has this character.

7. Page 6, lines 13-16: Mightn't this misrepresentation of stalling affect correlations where the influence might occur via the QBO in the lowermost stratosphere (i. e., at or just above the tropopause).

8. Page 10, line 10: Throughout this paper, the authors are careful to distinguish between correlations and cause-effect mechanisms. This is an exception.

9. Page 11, line 13: This presupposes that only one of the possible mechanisms is in play.

10. Page 12, line 16: Doesn't their figure 2 only show statistical significance in January?

11. Page 14, lines 21-24:  It's good that this is mentioned here, but I don't think this mechanism gets enough mention in this paper.  Given evidence for QBO influencing tropical precipitation, this likely affects Rossby wave trains that connect the tropics and extratropics.  Indeed, Ho et al. (2009) present evidence for such a wave train influencing typhoon tracks.  This should be mentioned on page 17, lines 15-19.
12. Page 16, line 16:  Liess and Geller (2012) also examined weather states from ISCCP that characterize active and mature deep convection.
13. Again, this paper would benefit from a summery table of QBO effects, levels of QBO most highly correlated, and possible mechanisms.

**Editorial Comments**

1. Page 1, line 20:  Insert "respectively" at end of line.
2. Page 2, line 14:  "mechanism" -> "mechanisms"  "is" -> "are"
3. Page 2, line 24:  "may" -> "seems to"
4. Page 7, lines 27-31:  This is a repetition of what is said near the top of this page.
5. Page 8, line 11:  Incomplete statement in parentheses.
6. Page 10, line 15:  The word "amended" seems odd.  Perhaps altered?
7. Page 10, line 23:   Only reduces it a bit.
8. Page 11, line 32:  Figure 5 doesn't show results including the polar vortex term.
9. Page 12, line 17:  Figure 76?
10. Page 16, lines 19-21:  Still another repetition.

---

## Referee Comment (RC2) · Anonymous Referee #2 · 10 Jan 2018

Review of "Surface Impacts of the Quasi Biennial Oscillation," By Lesley J. Gray, James A. Anstey, Yoshio Kawatani, Hua Lu, Scott Osprey, and Verena Schenzinger

The authors document the variability in zonal wind, precipitation, and SLP outside of the tropical stratosphere that are associated with the QBO. They do so by performing multivariate linear regression analysis of the QBO with the analyzed fields, and by testing the sensitivity to including a polar vortex index in their MLR, they are able to assess whether a given connection occurs predominantly through the vortex or through an alternate impact of the QBO. They find a QBO influence on the vortex and subsequently on the troposphere in January. In addition to this pathway, they also find a second significant signal in the North Pacific in February/March, and a third significant signal in the east North Atlantic and the Pacific unrelated to vortex variability in December. A fourth pathway is between the QBO and tropical convection/precipitation.

[Figure]

This paper could eventually become a valuable contribution to the field, but there are some major issues that need to be addressed as detailed below. Briefly, the authors need to convince the reader that the responses seen are not due to aliasing of SST variability (that may or may not have anything to do with ENSO). Furthermore, the assumption of nonlinearity inherent to MLR is somewhat suspect given the results of previous compositing studies.

General comments:

1. The authors need to nail down exactly why their precipitation response is so different from previous work (e.g. Leiss and Geller). I see three possibilities. First, there is a tendency, especially since 1979, for WQBO at 50hPa to occur simultaneous with El Nino (see Garfinkel and Hartmann 2007 and Leiss and Geller 2012). Both of these studies took a compositing approach to removing possible contamination of the ENSO signal from the QBO signal, and it is possible that the present study is aliasing an influence from SST variability (see general comments 3 and 4). Another possible difference between Leiss and Geller 2012 and the present study is the period examined: Leiss and Geller 2012 end their analysis in 2011 while the present study continues to 2015. If the authors end their analysis in 2011 (to match Leiss and Geller 2012), are the results more similar? A third possibility is the choice of the QBO index used (i.e. EOF vs pressure level based). Considering the authors already compare an EOF approach to a pressure level approach for SLP in a supplemental figure 5, I suggest they create a similar figure but for precipitation.

2. I suggest that the authors compare their precipitation patterns to those simulated in Garfinkel and Hartmann 2011 (part 2 using WACCM). Their various experiments have the same SSTs, and hence any differences in precipitation must arise via the QBO and not via aliasing of SSTs. I also note that GH11 conclude that the springtime SLP and wind anomalies in the North Pacific are not due to convection but rather due to the QBO affecting extratropical eddies directly (relevant to page 19 line 2). This paper should also be added to the list on p.3 line 11. Finally, this paper is also relevant to p. 10 line

31, as this paper proposes an answer to the question the authors raise; specifically, GH11 argue that the North Pacific circulation is more sensitive to external forcings in spring as compared to midwinter.

3. A single SST index cannot characterize the possible association between SSTs and the QBO. If one simply regresses (or composites) SST anomalies during different QBO phases (Huang et al 2012; Hu et al 2012), differences will appear in more than just the Nino3.4 region. Any apparent association between SSTs and the QBO is likely by chance, but the SST anomalies that necessarily will be present for any specific QBO phase could contribute to the extratropical response. I recommend that the authors create a figure analogous to their figure 8 but for SST anomalies. Such a figure could then be used to interpret the precipitation anomalies in figure 8, 9, and 10 (and also the SLP anomalies in the North Pacific in figure 6 and 7 which the authors relate to tropical convection). Specifically, are these convection and SLP anomalies just driven by SST anomalies that are present during these specific QBO phases, or can the convection and SLP anomalies be truly linked to the QBO?

4. The authors include an ENSO index in their MLR with the presumed intention that teleconnections of ENSO can be isolated from that of the QBO. However, MLR cannot account for nonlinear influences of ENSO on the remote response to the QBO. Similarly, MLR cannot account for nonlinear influences of the solar cycle on the remote response to the QBO. Previous work has indicated that such nonlinearities exist (Garfinkel and Hartmann 2007, Wei et al 2007, Labitzke 1987, Labitzke 2005). I suggest that the authors perform their MLR separately for El Nino years and La Nina years; the connection between the QBO and the vortex should be mainly present during La Nina if Wei et al and Garfinkel and Hartmann 2007 are indeed correct. Similarly I suggest that the authors perform their MLR separately for solar max years and solar min years; the connection between the QBO and the vortex should be reversed between these two assuming the Labitzke effects are robust.

Relatively minor comments 1, The authors seem convinced that an EOF approach

to characterizing the QBO is clearly better than a pressure level approach, but have failed to convince me. It is true that an EOF approach can characterize variability at different levels simultaneously better than a single level. However given the strong relationship between zonal wind anomalies at a given pressure level with any other pressure level (except in 2016, which was tough on EOF methods as well), this effect can be described by both EOF and pressure level methodologies. EOF methodologies have a downside: they are less intuitive and the specific choice on what phase angle corresponds to which wind profile is a subjective decision that can differ among studies. This can lead to confusion: p.7 line 23 states the relationship between the phases opposite to what is stated in the rest of the paper. As p.7 is the first time this relationship is stated, I wrote it down, only to become completely confused as I read the paper in my first pass. Eventually I figured out that p.7 line 23 was incorrect, but the fact that such a typo could occur in the first place illustrates the danger in using an EOF approach.

In my view, the main advantage of the EOF method is that it allows one to characterize (in principle at least) an infinite number of QBO phases. In the notation of the authors, one can create a QBO index for +60degrees, +30, 0, -20, -31.4159265359 degrees, etc. This allows for more flexibility in formulating the MLR. In contrast radiosonde data is available on some 7 pressure levels, and hence there are limitations on how once can test different phases. The authors don't mention this advantage, but I suggest they do so.

My intuition is that at the end of the day, it doesn't really matter which approach one follows, as is evidenced by their supplemental figure 5.

Technical comments p.2 line 8 There are earlier papers than that of Richter et al showing this (eg. Manzini et al 2006).

p.8 line 11 "hence our choice to show results at ****"

p 9 line 20 What is the linear correlation of the polar vortex index and the QBO? The correlation will certainly depend on the precise phase angle chosen, but as the authors

find a similar Holton-Tan effect for a wide range of phases I suspect the correlation will be similarly insensitive.

p. 10 The sentences starting on lines 5 and 8 seem to contradict as currently written. Please clarify.

Garfinkel, C. I., & Hartmann, D. L. (2007). Effects of the El Niño–Southern Oscillation and the Quasi‐Biennial Oscillation on polar temperatures in the stratosphere. Journal of Geophysical Research: Atmospheres, 112(D19).

Garfinkel, C. I., & Hartmann, D. L. (2011). The influence of the quasi-biennial oscillation on the troposphere in winter in a hierarchy of models. Part II: Perpetual winter WACCM runs. Journal of the Atmospheric Sciences, 68(9), 2026-2041.

Hu, Zeng-Zhen, Bohua Huang, James L. Kinter, Zhaohua Wu, and Arun Kumar. "Connection of the stratospheric QBO with global atmospheric general circulation and tropical SST. Part II: interdecadal variations." Climate dynamics 38, no. 1-2 (2012): 25-43.

Huang, Bohua, Zeng-Zhen Hu, James L. Kinter, Zhaohua Wu, and Arun Kumar. "Connection of stratospheric QBO with global atmospheric general circulation and tropical SST. Part I: methodology and composite life cycle." Climate dynamics 38, no. 1-2 (2012): 1-23.

Labitzke, Karin. "Sunspots, the QBO, and the stratospheric temperature in the north polar region." Geophysical Research Letters 14, no. 5 (1987): 535-537.

Labitzke, Karin. "On the solar cycle–QBO relationship: a summary." Journal of Atmospheric and Solar-Terrestrial Physics 67, no. 1 (2005): 45-54.

Wei, Ke, Wen Chen, and Ronghui Huang. "Association of tropical Pacific sea surface temperatures with the stratospheric Holton‐Tan Oscillation in the Northern Hemisphere winter." Geophysical Research Letters 34, no. 16 (2007).

---

## Author Comment (AC1) · 16 Mar 2018

**General response to both reviewers**

*We wish to thank you for very helpful and thoughtful reviews. We very much appreciate the time it has taken. As you will see, the paper has been substantially revised, and (we hope you agree) considerably improved. A large number of fairly minor changes to the text have been made to improve clarity; the major changes are highlighted in red in the attached manuscript.*

*The main changes are:*

1. *Prompted by reviewer 2, we have made a change of approach. We now employ a definition of the QBO index based on equatorial winds at selected pressure levels as the primary approach, and only use the EOF-based approach as a comparison. This provides the reader with a better assessment of whether the more complex EOF approach provides any added information or insight.*

2. *Prompted by reviewer 2, we have performed a much more extensive analysis of the precipitation signals. We have repeated the composite analysis of Leiss and Geller (2012) for the GPCC observational dataset and then, step by step we introduce the regression analysis approach and then perform identical analyses with the ERA dataset. We show by doing this that our results are compatible with the earlier results of Leiss and Geller.*

3. *As suggested by reviewer 1, we have included a schematic (now figure 1) showing the various possible pathways for QBO influence at the surface. We believe this, plus extensive reworking of the text to make it simpler to follow, will have achieved what is requested (we seriously considered a table, as suggested by the reviewer, but found this was rather cumbersome and hopefully a schematic serves the purpose better.*

**Specific responses to Review 1**

This is a very interesting, and valuable, paper. While many previous papers have looked at the influence of the stratospheric quasi-biennial oscillation (QBO) on various aspects of atmospheric behavior, such as on stratospheric jet structure, surface weather, and tropical precipitation, this paper examines all of these in a holistic manner. Furthermore, it seeks to shed some light on the possible mechanisms whereby the QBO may exert its influence. It does this by using multivariate linear regression analysis of the QBO with the analyzed fields. It includes other influences in this regression analysis, such as stratospheric volcanic aerosol abundance, ENSO, solar activity, and a long-term trend. Other interesting aspects to their analysis is that, rather than examining time series of the influence of the QBO winds at individual levels, they utilize time series of the first two QBO empirical orthogonal functions (EOFs) between 70 and 10 hPa. Noting that EOF1 maximizes in the upper part of this height range, while EOF2 maximizes in the lower portion, they make conclusions about which height range of the QBO influences various atmospheric fields.

There are a number of different pathways by which the QBO may influence atmospheric behavior. The QBO variation in equatorial winds modulates the winter waveguide for extratropical planetary waves, which can affect the strength of the polar vortex and stratospheric polar temperatures. The authors call this the "polar route." The QBO modulation of stratospheric equatorial winds is accompanied by meridional circulation, as is required by the thermal wind relationship. This, in turn, modulates equatorial stratospheric temperatures all the way down to the region below the tropical tropopause. This modulates tropical rainfall and tropical deep convection. This has been hypothesized in various observational analysis papers, and has also been shown in cloud-resolving modeling studies. The authors call this the "tropical route." The QBO equatorial meridional circulation has equatorial upwelling that decreases tropical temperatures in easterly shear conditions, while the return circulation increases temperatures through downwelling in the subtropics. This

affects the equatorial-subtropical temperature gradient, which in turn affects the wind shears in that region, which affects baroclinic waves and planetary waves. The authors call this the "subtropical route."

As mentioned previously, many previous authors have looked at QBO influences on the atmospheric circulation, but this paper differs from these in various ways. One is in the length of the data record being analyzed. The zonal wind a fields they analyze is the combined ERA-40 and ERA-Interim analyses that, together, extend from 1958-2016, while the precipitation is only the ERA-Interim analysis (1979-2016). The mean sea level pressure fields are from the Hadley Center for the period 1958 onward.

One interesting feature of their analysis is when they include both QBO and polar vortex indices in their regression analyses. This serves to separate effects where the QBO influence is through the "polar route," since QBO influences that remain after including the polar vortex index as a separate term in the regression analysis, would not likely involve the "polar route."

One shortcoming of this paper is its identification of QBO influences on so many aspects of the atmospheric circulation, that it is difficult to recall all of them. I suggest a summary table that includes all of the identified variables they identify as being influenced by the QBO along with their conclusions about the height range of the QBO that they identify as being key and also which of the three routes they believe to be most likely responsible.

*Please see the general response above, concerning the addition of a new schematic (new figure 1).*

These are my global comments. In the following, I make my more specific comments.

**Specific Comments**

1. Page 1, Lines 27-29: Given the fact that both QBO and solar indices are used in the authors' regression, I find it odd that Labitzke's work is not mentioned here. The Holton Tan mechanism seems to be opposite, depending on the phase of the solar cycle. This is particularly so since
several of the authors have written on this subject. Also, Labitzke et al. (2006) is in the reference list, but I can't find it in the text.

*Thank you for pointing this out – the reference is in there but somehow the text went missing. It is now inserted in section 1.*

2. Page 3, line 29: It seems to me that the reason why previous authors focus on the QBO at 40-50 hPa is that is where the amplitude of the QBO is maximum. That should probably be mentioned.

*Yes, 40-50 hPa is the height of the maximum amplitude of the lower stratospheric part of the QBO, but taking a look at the QBO amplitude diagnostics shown in Pascoe et al. (see their figure 4a) the actual maximum amplitude of the QBO lies higher up at 20 hPa – the 40-50 hPa level has probably been used traditionally because data from those levels were more freely available back in the 1980s when the Holton-Tan studies were performed, and mechanisms centred on the role of the lower stratosphere. However, inclusion of this discussion would be a distraction to the central theme of the paper and so we have left it out for brevity.*

3. Page 4, line 32: A reference is needed for this statement.

*This sentence has now been removed, since in the revised manuscript we have analysed data from the whole period.*

4. Page 5, line 4: It's interesting that 615 stations have provided data for more than 100 years. It doesn't seem that data for all those years are used in this paper though.
*Yes, this is true - the text has been modified to reflect this.*

5. Page 5, line 24: I think that the statement "that these indices are independent of one another" is too strong. Certainly, this is contradicted by statements in Garfinkel and Hartmann (2007), Salby (1996), and Taguchi (2010). It is sufficient to say that the inclusion/exclusion tests
were done.
*Yes, agreed, the text has been removed.*

6. Page 6, lines 4-11: Both EOFs have opposite signs at different altitudes. This is contrary to the implication that only EOF1 has this character.
*We have inserted the text 'In the height region 15-70 hPa' so there is no confusion.*

7. Page 6, lines 13-16: Mightn't this misrepresentation of stalling affect correlations where the influence might occur via the QBO in the lowermost stratosphere (i. e., at or just above the tropopause).
*Yes, thank you for raising this is as a possibility. In the revised manuscript we have changed our approach and show results for QBO defined at selected pressure surfaces as the main figures, and results from the EOF analysis as supporting figures. This has allowed a much better comparison of the two approaches (pressure level index versus EOF-based index) and we do not find substantial differences between them. We have added the following text: 'but comparison of results using the standard pressure level definition of the QBO index and the EOF-based definition do not show substantial differences, suggesting that this smoothing does not significantly affect the results'.*

8. Page 10, line 10: Throughout this paper, the authors are careful to distinguish between correlations and cause-effect mechanisms. This is an exception.
*Agreed – the text has been removed.*

9. Page 11, line 13: This presupposes that only one of the possible mechanisms is in play.
*Agreed – the text has been changed to 'one possible explanation'.*

10. Page 12, line 16: Doesn't their figure 2 only show statistical significance in January?
*Their figure only shows January (with statistical significance) – we only refer to these papers to show that this feature had already been identified.*

11. Page 14, lines 21-24: It's good that this is mentioned here, but I don't think this mechanism gets enough mention in this paper. Given evidence for QBO influencing tropical precipitation, this likely affects Rossby wave trains that connect the tropics and extratropics. Indeed, Ho et al. (2009) present evidence for such a wave train influencing typhoon tracks. This should be mentioned on page 17, lines 15-19.
*Yes, agreed – we have added to the page 17 text specifically reiterating this: 'an influence of the QBO on deep convection which can influence both the sub-tropical jet and also the mid-latitudes via a modulation of the tropical source region of large-scale planetary waves that propagate into the mid-latitudes (the equatorial route)'.*

12. Page 16, line 16: Liess and Geller (2012) also examined weather states from ISCCP that characterize active and mature deep convection.

*Comment refers to page 15, line 16? We mention only the composite analysis that Leiss and Geller performed because we have not carried out an analysis of active/mature convection and so cannot state that our results agree with that aspect of their study. In view of the review comments, we have repeated the GPCP composite analysis of Leiss and Geller to show more explicitly how our results compare with those of Leiss and Geller.*

13. Again, this paper would benefit from a summery table of QBO effects, levels of QBO most highly correlated, and possible mechanisms.

*Please see general response.*

**Editorial Comments**
1. Page 1, line 20: Insert "respectively" at end of line.

*Done*

2. Page 2, line 14: "mechanism" -> "mechanisms" "is" -> "are"

*Done*

3. Page 2, line 24: "may" -> "seems to"

*Done*

4. Page 7, lines 27-31: This is a repetition of what is said near the top of this page.

*Text has been removed*

5. Page 8, line 11: Incomplete statement in parentheses.

*Text has been removed*

6. Page 10, line 15: The word "amended" seems odd. Perhaps altered?

*Done*

7. Page 10, line 23: Only reduces it a bit.

*Text (and figure) has been amended to describe results for QBO at specified pressure levels.*

8. Page 11, line 32: Figure 5 doesn't show results including the polar vortex term.

*Yes, figure 5 was identical to figure 3 but with a linear scale, so it did include the vortex term – to avoid this misunderstanding we have amended the figure caption.*

9. Page 12, line 17: Figure 76?

*Amended - this should have been S6*

10. Page 16, lines 19-21: Still another repetition.

*Text has been removed.*

**Review 2**
The authors document the variability in zonal wind, precipitation, and SLP outside of the tropical stratosphere that are associated with the QBO. They do so by performing multivariate linear regression analysis of the QBO with the analyzed fields, and by testing the sensitivity to including a polar vortex index in their MLR, they are able to assess whether a given connection occurs predominantly through the vortex or through an alternate impact of the QBO. They find a QBO influence on the vortex and subsequently on the troposphere in January. In addition to this pathway, they also find a second significant signal in the North Pacific in February/March, and a third significant signal in the east North Atlantic and the Pacific unrelated to vortex variability in December. A fourth pathway is between the QBO and tropical convection/precipitation.

This paper could eventually become a valuable contribution to the field, but there are some major issues that need to be addressed as detailed below. Briefly, the authors need to

convince the reader that the responses seen are not due to aliasing of SST variability (that may or may not have anything to do with ENSO). Furthermore, the assumption of nonlinearity inherent to MLR is somewhat suspect given the results of previous compositing studies.

**General comments:**

1. The authors need to nail down exactly why their precipitation response is so different from previous work (e.g. Leiss and Geller).

*Yes, we agree that we need to nail down exactly where the differences in precipitation results come from. We have made some substantial changes to the paper, especially the precipitation analysis. Firstly, as suggested by the reviewer, we have repeated the precipitation analysis using standard pressure levels to define the QBO – in fact we have changed tack and show results from the standard pressure surface definition as the default, unless we can learn more from the EOF approach. So, for example, the zonal wind and mslp results are now shown using pressure surfacedefinition whereas the precipitation results show results from both approaches. In addition, we have accessed the GPCC dataset (essentially an updated version of that employed by Leiss and Geller 2012). We repeated their composite analysis and then performed a regression analysis on it using both standard pressure levels and the EOF-approach to define the QBO. We have done this for the annual-means (as in their paper) as well as the individual months. (We have also changed our colour scales to make it easier to compare with their results). Step by step, the comparisons with Leiss and Geller do not throw up any notable contradictions, and indeed it is not obvious now that our results are inconsistent with theirs (which is reassuring). We address specific comments more fully below.*

I see three possibilities. First, there is a tendency, especially since 1979, for WQBO at 50hPa to occur simultaneous with El Nino (see Garfinkel and Hartmann 2007 and Leiss and Geller 2012). Both of these studies took a compositing approach to removing possible contamination of the ENSO signal from the QBO signal, and it is possible that the present study is aliasing an influence from SST variability (see general comments 3 and 4).

*See response above – we do not believe there is aliasing with ENSO in our results; we had already done lots of sensitivity test e.g. check whether the ENSO signal looks sensible and whether the QBO index changed if it was removed etc). We have now shown step-by-step how we can get from GPCP composites through to ERA regression results with consistency at all the steps. Simply removing years with large ENSO anomalies from the compositing is not obviously better than representing those years with an ENSO index in the regression analysis (i.e. the years still need to be identified using a similar criterion, which is usually the ENSO3.4 index) – and in removing years from the composites the number of data points is substantially reduced, which is not the case with regression analysis.*

Another possible difference between Leiss and Geller 2012 and the present study is the period examined: Leiss and Geller 2012 end their analysis in 2011 while the present study continues to 2015. If the authors end their analysis in 2011 (to match Leiss and Geller 2012), are the results more similar?

*Yes, this contributes to small changes, but does not substantially change the results.*

A third possibility is the choice of the QBO index used (i.e. EOF vs pressure level based). Considering the authors already compare an EOF approach to a pressure level approach for SLP in a supplemental figure 5, I suggest they create a similar figure but for precipitation.

*Yes, agreed, we have changed tack and show the pressure level approach first, before moving on the EOF approach.*

2. I suggest that the authors compare their precipitation patterns to those simulated in Garfinkel and Hartmann 2011 (part 2 using WACCM). Their various experiments have the same SSTs, and hence any differences in precipitation must arise via the QBO and not via aliasing of SSTs. I also note that GH11 conclude that the springtime SLP and wind anomalies in the North Pacific are not due to convection but rather due to the QBO affecting extratropical eddies directly (relevant to page 19 line 2). This paper should also be added to the list on p.3 line 11. Finally, this paper is also relevant to p. 10 line 31, as this paper proposes an answer to the question the authors raise; specifically, GH11 argue that the North Pacific circulation is more sensitive to external forcings in spring as compared to midwinter.

*Thank you – text has been added in all the suggested sections.*

3. A single SST index cannot characterize the possible association between SSTs and the QBO. If one simply regresses (or composites) SST anomalies during different QBO phases (Huang et al 2012; Hu et al 2012), differences will appear in more than just the Nino3.4 region. Any apparent association between SSTs and the QBO is likely by chance, but the SST anomalies that necessarily will be present for any specific QBO phase could contribute to the extratropical response. I recommend that the authors create a figure analogous to their figure 8 but for SST anomalies. Such a figure could then be used to interpret the precipitation anomalies in figure 8, 9, and 10 (and also the SLP anomalies in the North Pacific in figure 6 and 7 which the authors relate to tropical convection). Specifically, are these convection and SLP anomalies just driven by SST anomalies that are present during these specific QBO phases, or can the convection and SLP anomalies be truly linked to the QBO?

*The possible aliasing of the QBO response with SST anomalies is a really interesting avenue for further exploration, and the studies of Huang et al and Hu et al are illuminating. However, this would significantly add to the complexity of the analysis and its interpretation, which is already rather over-length. Even comparison in the text with the results of Huang and Hu is rather challenging, especially because of the different methodologies employed. Instead, we have noted this as a caveat at the end of the summary sections, in addition to other caveats that require further exploration.*

4. The authors include an ENSO index in their MLR with the presumed intention that teleconnections of ENSO can be isolated from that of the QBO. However, MLR cannot account for nonlinear influences of ENSO on the remote response to the QBO. Similarly, MLR cannot account for nonlinear influences of the solar cycle on the remote response to the QBO. Previous work has indicated that such nonlinearities exist (Garfinkel and Hartmann 2007, Wei et al 2007, Labitzke 1987, Labitzke 2005). I suggest that the authors perform their MLR separately for El Nino years and La Nina years; the connection between the QBO and the vortex should be mainly present during La Nina if Wei et al and Garfinkel and Hartmann 2007 are indeed correct. Similarly I suggest that the authors perform their MLR separately for solar max years and solar min years; the connection between the QBO and the vortex should be reversed between these two assuming the Labitzke effects are robust.

*Yes, agreed, there are significant non-linearities, and several of our authors have already explored this in the past, but further analysis along the suggested lines would require a separate paper. We have noted the issue of non-linearities in an additional final paragraph to the summary section, listing a number of caveats.*

Relatively minor comments 1, The authors seem convinced that an EOF approach to characterizing the QBO is clearly better than a pressure level approach, but have failed to convince me. It is true that an EOF approach can characterize variability at different levels simultaneously better than a single level. However given the strong relationship between zonal wind anomalies at a given pressure level with any other pressure level (except in 2016, which was tough on EOF methods as well), this effect can be described by both EOF and pressure level methodologies. EOF methodologies have a downside: they are less intuitive and the specific choice on what phase angle corresponds to which wind profile is a subjective decision that can differ among studies. This can lead to confusion: p.7 line 23 states the relationship between the phases opposite to what is stated in the rest of the paper. As p.7 is the first time this relationship is stated, I wrote it down, only to become completely confused as I read the paper in my first pass. Eventually I figured out that p.7 line 23 was incorrect, but the fact that such a typo could occur in the first place illustrates the danger in using an EOF approach.

In my view, the main advantage of the EOF method is that it allows one to characterize (in principle at least) an infinite number of QBO phases. In the notation of the authors, one can create a QBO index for +60degrees, +30, 0, -20, -31.4159265359 degrees, etc. This allows for more flexibility in formulating the MLR. In contrast radiosonde data is available on some 7 pressure levels, and hence there are limitations on how once can test different phases. The authors don't mention this advantage, but I suggest they do so.

My intuition is that at the end of the day, it doesn't really matter which approach one follows, as is evidenced by their supplemental figure 5.

*As already mentioned, we have changed tack, and present results from the more intuitive pressure level approach as default, and only show the EOF results when they add more information. We have also added some sentences to spell out more clearly the benefits of the EOF approach. We also apologise for the rather unfortunate typo on p7 line 23, which has now been corrected.*

**Technical comments**

p.2 line 8 There are earlier papers than that of Richter et al showing this (eg. Manzini et al 2006).

*We try here to give a paper with an overview, and pointers to previous references, which is why we chose a later paper rather than the first ever study – the reference has been changed to Hansen et al. 2016, to do this better.*

p.8 line 11 "hence our choice to show results at ****"

*Thank you – this text has now been removed*

p 9 line 20 What is the linear correlation of the polar vortex index and the QBO? The correlation will certainly depend on the precise phase angle chosen, but as the authors find a similar Holton-Tan effect for a wide range of phases I suspect the correlation will be similarly insensitive.

*The correlation between QBO and polar vortex has been studied by many authors, giving something like 0.4-0.5 correlation; there is also some variation depending on the period examined. Yes, agreed, the correlation is likely to be similarly insensitive.*

p. 10 The sentences starting on lines 5 and 8 seem to contradict as currently written. Please clarify.

*These sentences have been removed during revision of the text.*

Garfinkel, C. I., & Hartmann, D. L. (2007). Effects of the El Niño–Southern Oscillation and the QuasiâRˇBiennial Oscillation on polar temperatures in the stratosphere. Journal of Geophysical Research: Atmospheres, 112(D19).

Garfinkel, C. I., & Hartmann, D. L. (2011). The influence of the quasi-biennial oscillation on the troposphere in winter in a hierarchy of models. Part II: Perpetual winter WACCM runs. Journal of the Atmospheric Sciences, 68(9), 2026-2041.

Hu, Zeng-Zhen, Bohua Huang, James L. Kinter, Zhaohua Wu, and Arun Kumar. "Connection of the stratospheric QBO with global atmospheric general circulation and trop- ical SST. Part II: interdecadal variations." Climate dynamics 38, no. 1-2 (2012): 25-43.

Huang, Bohua, Zeng-Zhen Hu, James L. Kinter, Zhaohua Wu, and Arun Kumar. "Connection of stratospheric QBO with global atmospheric general circulation and tropical SST. Part I: methodology and composite life cycle." Climate dynamics 38, no. 1-2 (2012): 1-23.

Labitzke, Karin. "Sunspots, the QBO, and the stratospheric temperature in the north polar region." Geophysical Research Letters 14, no. 5 (1987): 535-537.

Labitzke, Karin. "On the solar cycle–QBO relationship: a summary." Journal of Atmospheric and Solar-Terrestrial Physics 67, no. 1 (2005): 45-54.

Wei, Ke, Wen Chen, and Ronghui Huang. "Association of tropical Pacific sea surface temperatures with the stratospheric HoltonâRˇTan Oscillation in the Northern Hemisphere winter." Geophysical Research Letters 34, no. 16 (2007).

---

## Author Response (AR2)

Dear Editor and Reviewers

Thank you for your reviews – we very much appreciated the time and attention that has been provided, resulting in a much improved manuscript. Detailed responses are provided below; the amended text is highlighted in the revised manuscript.

Yours Sincerely
Lesley Gray

**Response to reviewer 1:**

In the abstract on line 9, the authors say that the ITCZ precipitation response may be particularly sensitive to the vertical wind shear. I agree with this, but I think what is most
important here is the tropical tropopause temperature. This is what modeling shows. Of course, the wind shear and temperature are related by the thermal wind relationship. I recommend that the authors indicate that what is likely is that the tropopause temperatures are what are important. It would be difficult for me to imagine a physical mechanism directly coupling deep convection to 70 hPa wind shear.
*Thank you – some additional text has been added to the abstract.*

On page 3, line 30, I recommend deleting the word "equatorial." Both small-scale gravity waves and larger scale equatorial waves are important in giving rise to the QBO. to me, leaving the word "equatorial" in here implies the authors think it is the larger scale waves that are most important.
*Done.*

Page 4. line 12. Space between "be" and "captured."
*Corrected.*

Page 6, discussion of figure S2. It looks to me that the QBO amplitudes are larger at high levels in the ERA results. Do the authors agree? If so, do they have an explanation?
*Yes, that's interesting, and I'm not sure why – I haven't added any text to this effect since it's outside the topic of the current study, but it would be worth pursuing further.*

Page 8, line 11. "highlight"
*Corrected.*

Page 9. line 19. I have difficulty seeing what the authors are referring to. Could they be more explicit in describing what they mean here?
*Text has been amended to make the discussion more explicit.*

Page 16, lines 20-26. Liess and Geller (2012) examined ISCCP Weather States for developing and mature convection, so their results agree with the authors'

conclusions. Perhaps, that should be indicated.
*Text has been added to mention this agreement.*

Page 17, line 8. Again, it should be made clear that indications are that it is the temperature in the tropopause region that is most important here.
*Text has been added, as recommended.*

**Response to Reviewer 2:**

Page 2, line 11 There are earlier papers than that of Hansen that address nonlinearities between the effect of the QBO and that of ENSO, such as Garfinkel and Hartmann 2007, Wei et al 2007, and Calvo et al 2009 (the first two using reanalysis and the latter using models).
*Additional references have been added, as suggested.*

Page 6 line 4: what precise formula is used to account for the reduction in degrees of freedom due to autocorrelation of the indices? (There are a few different I'm aware of, and for reproducibility the authors should state which they use.)
*Text has been clarified and an additional reference provided.*

Page 14 line 16: I think the authors intend to cite Garfinkel and Hartmann 2011 here, not Garfinkel et al 2012
*Corrected.*